# Bioinspired ruthenium-manganese-oxygen complex for biocatalytic and radiosensitization therapies to eradicate primary and metastatic tumors

Ruidan Li[1,7], Ting Wang[2,7], Shengdong Mu[3], Zhenyu Xing [2], Zhiying Ding[2], Qinlong Wen[1], Zhigong Wei[1], Xiaolin Wang [4], Mohsen Adeli[5], Shuang Li [2], Chong Cheng [2,6] ✉ & Xingchen Peng [1] ✉

Designing efficient, biocompatible radiation-sensitive materials to activate systemic immune responses can maximize tumoricidal effects against malignant tumors. Here, inspired by natural Mn-peroxidase, we propose the de novo design of the RuMn-oxygen complex (MnBTC-Ru) for biocatalytic and radiosensitization therapies to eradicate primary and metastatic tumors. Our results reveal that Mn-organic ligands can enhance the electron density of Ru clusters, thereby optimizing their binding to oxygen species and resulting in high reactive oxygen species and oxygen generation. Accordingly, MnBTC-Ru with radiation can enhance cell membrane and DNA damage, triggering apoptosis though oxidative damage, heightening radiosensitization, and activating CD8+ T cells. When combined with anti-PD-1 therapy, this synergistic approach generates robust systemic antitumor responses in female mice, promoting the abscopal effect and establishing enduring immune memory against tumors, thereby reducing recurrence and metastasis. This design presents superior biocatalytic and radiosensitizing properties, which may provide promising and practical bio-nanotechnology for future treatments on eradicating primary and metastatic tumors.

Cancer is a leading cause of death worldwide, responsible for millions of deaths each year and posing a significant challenge to global public health[1–4]. Radiotherapy (RT), a widely utilized cancer treatment strategy that generates reactive oxygen species (ROS) and induces double-strand DNA breaks, has demonstrated significant efficacy when combined with immune checkpoint inhibitors in treating refractory, recurrent, and metastatic tumors, marking a breakthrough in cancer treatment[5,6]. Nevertheless, factors such as radiation resistance, tumor heterogeneity, and the tumor microenvironment (TME) pose significant challenges to the clinical efficacy of RT-based immune checkpoint inhibitor therapy in certain solid tumors[7–10]. Recently, studies have shown that creating radiation-sensitive drugs or materials to stimulate both local and systemic immune responses can maximize the tumoricidal effects on primary, regional recurrence, and

[1]Department of Biotherapy, Cancer Center, West China Hospital, Sichuan University, Chengdu, China. [2]College of Polymer Science and Engineering, State Key Laboratory of Advanced Polymer Materials, Sichuan University, Chengdu, China. [3]College of Biological Engineering, Sichuan University of Science and Engineering, Yibin, China. [4]School of Pharmacy and State Key Laboratory of Quality Research in Chinese Medicine, Macau University of Science and Technology, Macao, China. [5]Institute of Chemistry and Biochemistry, Free University of Berlin, Berlin, Germany. [6]Department of Endodontics, State Key Laboratory of Oral Diseases, National Clinical Research Center for Oral Diseases, West China Hospital of Stomatology, Sichuan University, Chengdu, China. [7]These authors contributed equally: Ruidan Li, Ting Wang. ✉ e-mail: chong.cheng@scu.edu.cn; pxx2014@scu.edu.cn

metastatic tumors[11–13]. However, the currently developed radio-sensitizing materials (e.g., lanthanides, hafnium, and gold) are limited in their potential clinical translation due to their insufficient tumoricidal effects and low biodegradability, which may lead to chronic toxicities[14–16].

Designing next-generation radiosensitizing materials to meet clinical tumor RT treatments encounters two significant challenges: 1) developing highly biocompatible and biodegradable radiosensitizing materials that enhance the energy deposition of X-ray radiation and ROS production; 2) the enhancement of ROS and oxygen ($O_2$) generation to reverse the hypoxic TME condition, mitigate radio-resistance, reduce material dosage, and activate systemic antitumor responses, thereby maximizing the therapeutic efficacy and preventing metastasis and recurrence[17–20]. In recent years, artificial peroxidases that can generate ROS and $O_2$ in acidic and hydrogen peroxide ($H_2O_2$)-rich TME are gaining increased popularity[21–23]; for instance, metal oxides, metal hydroxides, and metal-coordinated carbon nanomaterials[21,24–26]. However, developing high-performance ROS- and $O_2$-generating artificial peroxidases remains challenging due to the complex multielectron reactions involving $H_2O_2$ molecules and oxygen radicals, intricate formation of intermediate bonds, and energy-intensive desorption of oxygen species[27–31]. Furthermore, most artificial peroxidases consist of abundant inorganic or metallic components, which raises concerns regarding biological safety as next-generation radiosensitizing materials[32–35].

Natural peroxidases, such as manganese (Mn)-peroxidase, which features Mn-oxygen complex site coordinated by organic ligands with spatial configuration, exhibit high ROS generation capabilities by facilitating the entry of reactants through distinctive three-dimensional spatial arrangements and modulating the redox reactions of the Mn sites via organic ligand coordination[36–38]. Recently, our research team reported that ruthenium (Ru), a relatively high atomic number metal belonging to the iron group, possesses good biocompatibility and unique biocatalytic properties—specifically, more $d$ electrons, sufficient unoccupied orbitals, and superior redox stability—that enable efficient $O_2$ generation through rapid electron transfer with $H_2O_2$ as the substrate[39–42]. Therefore, by mimicking the structural properties and Mn active sites of natural Mn-peroxidase, the development of artificial peroxidase materials that integrate Mn-oxygen complex with Ru species through organic ligand coordination strategies is anticipated to offer a biocompatible and biodegradable approach to overcome the challenges associated with unbalanced multielectron reactions involving ROS and $O_2$ cycling catalysis[43–45], while this has not been reported.

Here, drawing inspiration from the coordination and catalytic properties of natural Mn-peroxidase, we propose the de novo design of RuMn-oxygen complex (MnBTC-Ru) that contains catalytic sites formed by Ru clusters coordinated within the Mn-organic ligands for biocatalytic and radiosensitization therapies to eradicate primary and metastatic tumors (Fig. 1a). The core motivation behind this research stems from two key factors: 1) the bioinspired design of MnBTC-Ru showcases dual functional biocatalytic properties and effective radiosensitizing capabilities, swiftly increasing ROS and $O_2$ levels concurrently in TME condition to maximize the tumoricidal effects on primary tumor; and 2) the biocompatible and biodegradable Mn-organic coordination structures of MnBTC-Ru allow for potential applications for stimulating systemic antitumor responses to prevent metastatic tumors. Notably, our experimental and theoretical investigations reveal that the Mn-organic ligands play a critical role in increasing the electron density of Ru clusters, which optimizes the binding affinity of oxygen species and results in high ROS and $O_2$ generation capacities. Our findings have disclosed that the MnBTC-Ru can effectively enhance DNA damage and trigger robust apoptosis by promoting oxidative damage to the cell membrane, thereby increasing the sensitivity of cancer cells to X-ray irradiation. Accordingly, the MnBTC-Ru+RT treatment promotes a transformative response in the tumor's microenvironment, heightening radiosensitization and activating CD8+ T cells within the tumor. Furthermore, when combined with anti-PD-1 therapy, this synergetic approach leads to intensive systemic antitumor responses, fostering the abscopal effect and establishing enduring memory against tumors, which effectively prevents tumor recurrence and metastasis and significantly enhances treatment outcomes and long-term prognosis. Noteworthy, our design showcases biocatalytic and radiosensitizing properties, thus providing promising bio-nanotechnology with superior effects on eradicating primary and metastatic tumors, especially malignant tumors.

## Results

### Design and structure characterizations of MnBTC-Ru complex

In a typical synthetic process, the MnBTC-Ru complex, containing catalytic sites formed by Ru clusters coordinated with the Mn-organic ligands, is synthesized through a sequential reaction process. First, Mn-organic precursors (MnBTC) are created via a one-step hydrothermal method that combines Mn salt and 1,3,5-benzenetricarboxylic acid (BTC). Subsequently, Ru ions are introduced into the MnBTC through metal ion adsorption, exchange, and nucleation processes to form MnBTC-Ru (Fig. 1a). For comparisons, carbon-supported Ru sites without spatial organic ligands, denoted as C-Ru, are also synthesized following methods outlined in the Methods. Inspired by the inherent catalytic properties of Mn-organic ligands in natural Mn-peroxidase, it can be inferred that the developed biocatalytic MnBTC-Ru complex may offer several structural advantages (Fig. 1b). These advantages include: i) the ability of Mn-BTC ligands to donate electrons, ii) the facilitation of *OOH desorption by the electron-rich Ru clusters, and iii) the capacity of MnBTC-Ru with spatial organic ligands for efficient and cycling ROS. X-ray diffraction analysis confirms the successful synthesis of crystalline MnBTC support (Supplementary Fig. 1). The molecular structure of MnBTC-Ru is evaluated through Fourier transform infrared (FTIR) spectroscopy, revealing the presence of -OH and -COOH groups, which are consistent with those found in MnBTC (Fig. 1c).

Scanning electron microscopy (SEM) and transmission electron microscopy (TEM) images illustrate that MnBTC-Ru showcases well-preserved spherical structures with an average diameter of around 350 nm (Fig. 1d and Supplementary Figs. 2, 3). Dynamic light scattering analysis indicates that MnBTC-Ru maintains good dispersion in both phosphate-buffered saline (PBS) and mouse serum, with average hydrodynamic diameters of 578 and 525 nm, respectively (Supplementary Fig. 4). These values closely correspond to the inherent particle size of MnBTC-Ru, demonstrating favorable dispersibility under simulated physiological conditions and supporting its low potential for in vivo aggregation and related adverse effects. Furthermore, detailed crystal facets of Ru clusters within MnBTC-Ru are observed in high-resolution TEM images, offering views along the (100) and (110) planes that align precisely with facet distances (Fig. 1e). High-angle annular dark-field scanning transmission electron microscopy (HAADF-STEM) images illustrate the presence of Ru clusters (denoted by yellow arrows) with an average size of 1.08 nm (Fig. 1f, g). Subsequent atomic-resolution HAADF-STEM imaging reveals that the Ru clusters (marked by yellow circles) are surrounded by a few atomically dispersed Mn atoms (marked by green circles; Fig. 1h and Supplementary Fig. 5). Energy-dispersive spectroscopy (EDS) mappings and spectra demonstrate the existence of Ru, Mn, C, and O elements, with Ru clusters primarily distributed on the outer surface of the MnBTC spherical structure (Fig. 1i–k and Supplementary Fig. 6).

X-ray photoelectron spectroscopy (XPS) is employed to investigate the valence states and electronic configurations of the biocatalytic MnBTC-Ru complex (Supplementary Table 1). The high-resolution O 1$s$ spectra validate the presence of metal-O peaks in both MnBTC-Ru and MnBTC, thus verifying the formation of MnBTC (Fig. 2a). Mn 2$p$

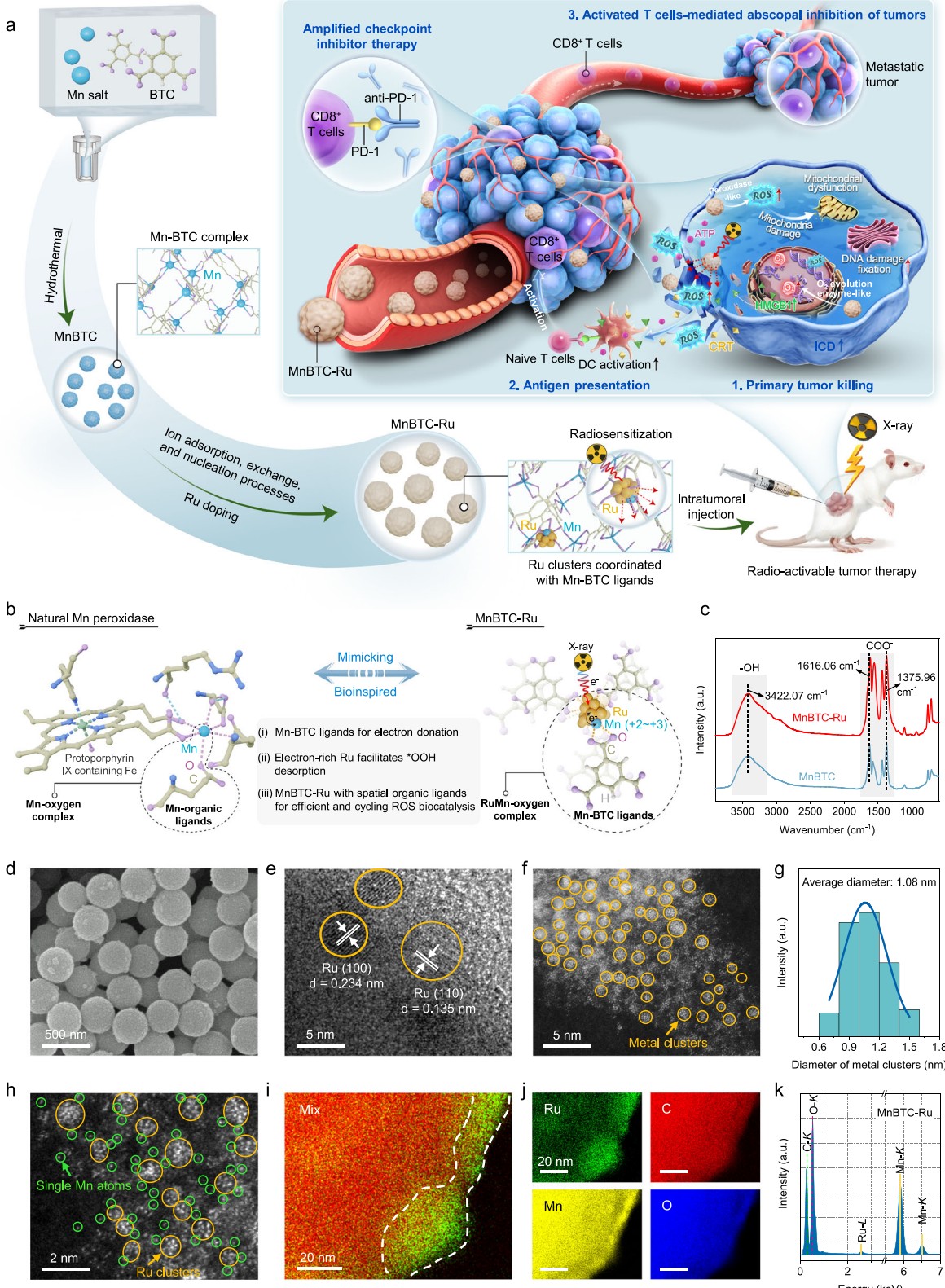

**Fig. 1 | Design and structure characterizations of MnBTC-Ru complex.**
**a** Schematic illustration of the synthesis of MnBTC-Ru complex and its role as a biocatalytic and radiosensitive agent that activates systemic antitumor response, facilitating the eradication of primary and metastatic tumors. **b** Natural Mn-peroxidase-inspired construction of RuMn-oxygen complex for ROS biocatalysis. **c** FTIR spectrum of MnBTC-Ru and MnBTC. **d** SEM, **e** TEM, and **f** HAADF-STEM images of MnBTC-Ru. **g** Diameter analysis of Ru clusters on MnBTC-Ru. **h** Atomic-resolution HAADF-STEM image of MnBTC-Ru. **i, j** EDS mapping and **k** EDS spectra of

MnBTC-Ru. Experiments were repeated independently (**d**–**f**, **h**–**j**) three times with similar results. In (**a**), ICD indicates immunogenic cell death, HMGB1 indicates high mobility group box-1, DC indicates dendritic cell, CRT indicates cell surface calreticulin, ATP indicates adenosine triphosphate. Atomic color coding in (**a**, **b**): Ru, yellow; Mn, blue; C, khaki; H, white; O, purple. In (**i**), Mix indicates mixture. In (**c**, **g**, **k**), a.u. indicates the arbitrary units. Source data are provided as a Source Data file.

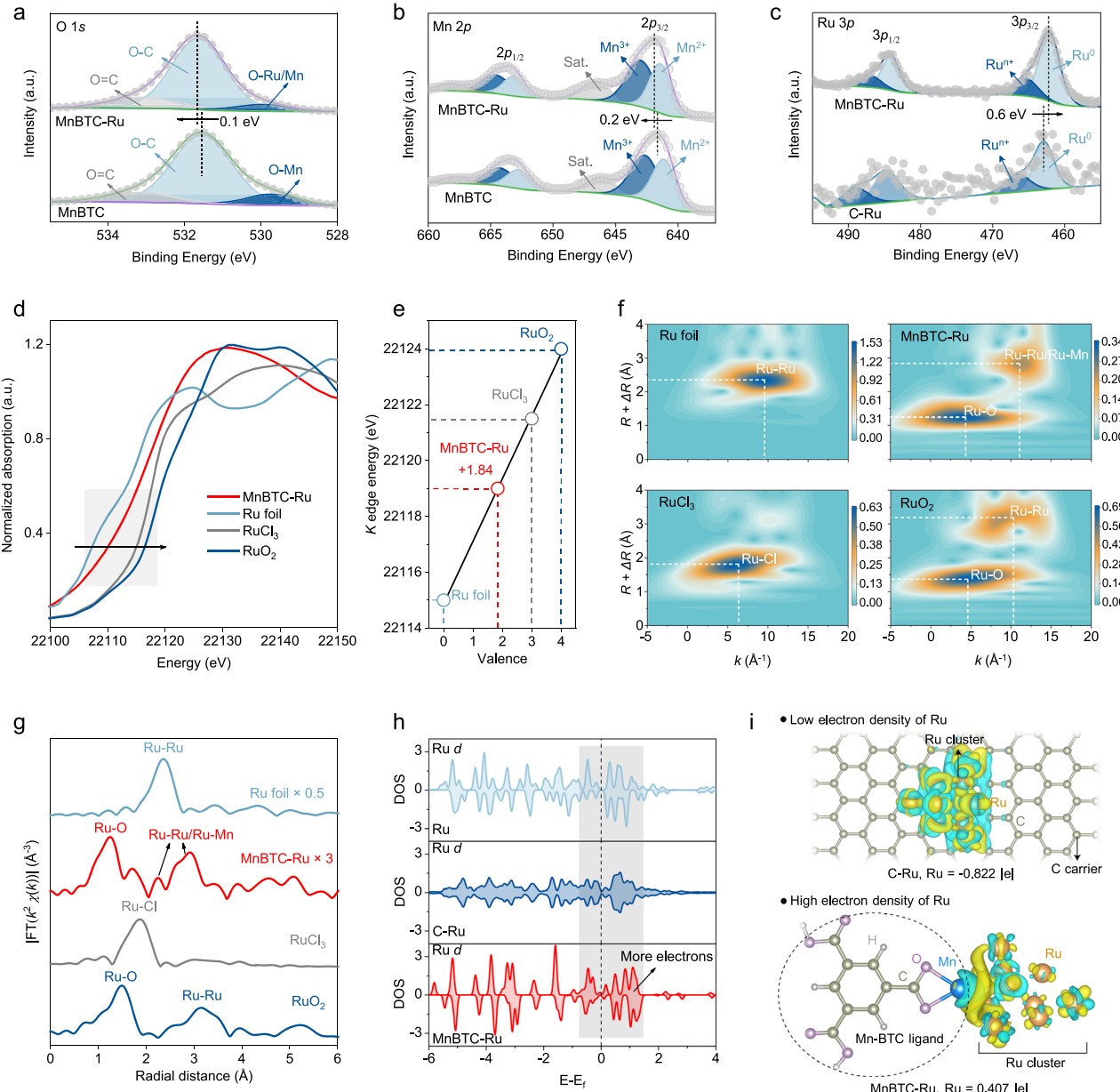

**Fig. 2 | Analysis of precise atomic coordination structures in MnBTC-Ru complex.** The high-resolution XPS of **a** O 1*s*, **b** Mn 2*p*, and **c** Ru 3*p* for different catalysts. **d** Ru *K*-edge XANES spectra of MnBTC-Ru and references (Ru foil, RuCl$_3$, and RuO$_2$). **e** Valence analysis of Ru species in MnBTC-Ru. **f** WT images of Ru *K*-edge EXAFS for different samples. Colour gradients have no units; the magnitude of the value indicates the intensity. **g** Ru *k$^2$*-weighted Fourier transform (FT) spectra in *R*-space. **h** DOS analysis of Ru 4*d* orbital of Ru particles, C-Ru, and MnBTC-Ru. **i** Differential charge density analysis of Ru centers (yellow and cyan represent charge accumulation and depletion, respectively, with a cutoff value of 0.002 e·Bohr$^{-3}$ for the density-difference isosurface). Atomic color coding in the structure: Ru, yellow; Mn, blue; C, khaki; H, white; and O, purple. In (**b**), Sat. indicates the satellite peak. In (**a**–**d**), a.u. indicates the arbitrary units. Source data are provided as a Source Data file.

analysis reveals the coexistence of Mn$^{2+}$ and Mn$^{3+}$ valence states within MnBTC-Ru, with MnO$_6$ centers coordinated exclusively to the oxygen atoms of BTC ligands exhibiting a +3 oxidation state, Mn species with unsaturated coordination exhibiting a +2 oxidation state, and electron transfer between the RuMn active centers inducing Mn to display valence states ranging from +2 to +3 (Fig. 2b and Supplementary Fig. 7)[46–48]. Notably, the O 1*s* and Mn 2*p* spectra of MnBTC-Ru show a positive core level shift compared to those of MnBTC. In contrast, the Ru$^0$ species in MnBTC-Ru exhibit a significant negative level shift of 0.6 eV in the binding energy of the 3*p$_{3/2}$* orbital when compared to C-Ru (Fig. 2c). These findings suggest that electrons are transferred from the electron-donating MnBTC to the Ru cluster sites. This

electron transfer may facilitate overcoming the multiple electron reactions of oxygen intermediates on the Ru clusters, thereby ensuring a rapid redox reaction. Furthermore, the high electronegativity of Ru species, characterized by a Pauling scale value of 2.2, is also beneficial to enhance the electron transfer process.

Subsequently, X-ray absorption near-edge structure (XANES) spectroscopy and extended X-ray absorption fine structure (EXAFS) spectroscopy are employed to elucidate the atomic coordination environments of the Ru centers within the biocatalytic MnBTC-Ru complex. Analysis of the Ru *K*-edge XANES spectra reveals that the pre-edge peak of MnBTC-Ru falls between Ru foil and RuCl$_3$, with the average valence state of Ru being +1.84, indicating the presence of

partially oxidized Ru clusters in the biocatalyst (Fig. 2d, e). Furthermore, the wavelet transform (WT) and Fourier transform of the $k^2$-weighted EXAFS spectra demonstrate the existence of Ru-O and Ru-Ru/Ru-Mn bonds in MnBTC-Ru, signifying the successful coordination of the Ru clusters with the Mn-BTC ligands (Fig. 2f, g). Subsequently, we employed density functional theory (DFT) to examine the influence of Mn-BTC ligands on the electronic structure of Ru clusters. To enhance computational efficiency, we employed a Ru cluster coordinated by a Mn-BTC ligand as a representative computational model. The introduction of both the carbon substrate and MnBTC has a significant impact on the density of states (DOS) of the Ru clusters (Fig. 2h). Notably, due to the electron-donating characteristics of MnBTC, the Ru sites (0.407 |e|) exhibit a higher electron density compared to those in C-Ru (-0.822 |e|; Fig. 2i). This increased electron density at the Ru sites enhances the capability for overcoming multi-electron redox reactions with oxygen species.

### Enzyme-mimetic ROS-biocatalytic activities

Following the characterization of the morphology and electronic structures of the synthesized MnBTC-Ru complex, particularly its Ru clusters coordinated within the Mn-BTC ligands, we systematically verify its enzyme-mimetic biocatalytic capabilities. We propose that the biocatalytic MnBTC-Ru complex exhibits dual functionality in generating both $O_2$ and ROS; meanwhile, it can produce ROS in significant quantities under X-ray radiation, thereby enhancing radiosensitization, reducing radiation resistance, and improving its overall antitumor efficacy compared to the control biocatalyst (C-Ru) (Fig. 3a).

Initially, the $O_2$-generation activity was investigated in the presence of $H_2O_2$ at pH = 6.5 to mimic the TME conditions, showcasing the superior performance of MnBTC-Ru, which achieves an $O_2$ concentration of 29.59 mg/L in 300 seconds (Fig. 3b). The peroxidase-like activity of biocatalysts was assessed using the 3,3′,5,5′-tetramethylbenzidine (TMB) assay, revealing that MnBTC-Ru displays rapid ROS generation capabilities compared to C-Ru and bare MnBTC (Fig. 3c and Supplementary Figs. 8, 9). Subsequent treatment of the metal sites with potassium thiocyanide (KSCN) can cause a notable reduction in the peroxidase-like activity of MnBTC-Ru, underscoring the essential role of Ru and Mn centers as active sites for ROS generation (Fig. 3d).

Thereafter, the kinetics of the peroxidase-mimetic process are thoroughly investigated. Essential parameters such as the catalytic Michaelis constant ($K_m$), maximum reaction velocity ($V_{max}$), and turnover number (TON, indicating the maximum substrate conversions per catalytic atom) are methodically determined utilizing Michaelis-Menten analysis (Supplementary Figs. 10, 11). In comparison to C-Ru, the MnBTC-Ru exhibits significantly higher $V_{max}$ (2.25 μM s$^{-1}$) and TON (71.60 × 10$^{-3}$ s$^{-1}$), pointing to more effective catalytic kinetics (Fig. 3e and Supplementary Table 2). Importantly, when contrasted with recently reported biocatalysts, the MnBTC-Ru showcases the highest TON values (Fig. 3f and Supplementary Table 3). The types of ROS generated by MnBTC-Ru in the presence of $H_2O_2$ are determined through free radical quenching experiments, verifying that $\bullet O_2^-$ and $^1O_2$ are the primary ROS products (Supplementary Fig. 12). The existence of $\bullet O_2^-$ and $^1O_2$ is additionally confirmed through the dihydroethidium (HE) and 9,10-diphenanthraquinone (DPA) probe, respectively (Fig. 3g, h). Subsequent use of electron paramagnetic resonance (EPR) spectroscopy, the o-phenylenediamine (OPD) colorimetric assay reveals the enhanced ROS production by the biocatalytic MnBTC-Ru complex in the presence of X-ray radiation (Fig. 3i, j and Supplementary Fig. 13). The Ru-Mn synergistic center, together with its cooperative interaction with RT, constitutes the principal mechanism driving ROS generation. In-situ FTIR spectroscopy has also been employed to examine the structural changes of the ROS intermediates during the peroxidase-like process, indicating the formation of $\bullet O_2^-$ and OOH intermediates (Fig. 3k, l)[49–51].

Considering that, during the catalytic process, the dynamic equilibrium between OOH adsorbed (*OOH) on the material surface and protons ($H^+$) in solution constitutes the central mechanism to the generation of $\bullet O_2^-$, and that the formation of $^1O_2$ similarly relies on the transformation of *OOH via the Russell mechanism[52–56], we, therefore, regard *OOH as an ideal and essential model for a deeper investigation into the intrinsic relationship between the structure of the Ru cluster coordinated with Mn-organic ligands and its catalytic properties through DFT calculations. As illustrated in Fig. 3m, n, the electronic interaction between Ru in MnBTC-Ru and *OOH is weaker compared to that in the C-Ru, a difference that can be quantitatively assessed by the calculated Bader charge. Additionally, DOS analysis reveals that MnBTC-Ru demonstrates reduced orbital overlap with *OOH, further supporting the conclusion that MnBTC-Ru is advantageous for *OOH desorption (Fig. 3o). Overall, the electronic structure modulation facilitated by MnBTC optimizes the binding strength of Ru cluster with *OOH, thereby enhancing its catalytic properties for the generation of ROS and $O_2$.

### MnBTC-Ru eradicates primary tumor and activates immunity

The efficient generation of ROS and $O_2$ is a critical factor for radiosensitizing biocatalysts to overcome radioresistance and reverse the suppressive TME[57,58]. Firstly, we evaluate the ROS and $O_2$ generation capacities of MnBTC-Ru in vitro, which directly or indirectly promotes tumor cell apoptosis induced by RT (Fig. 4a). ROS levels were assessed using the ROS probe, 2′,7′-dichlorodihydrofluorescein diacetate (DCFH-DA). The results show that tumor cells treated with RT+MnBTC-Ru exhibit the highest ROS generation, as indicated by fluorescence imaging and quantitative flow cytometry (Fig. 4b, c). The ROS generated by MnBTC-Ru can lead to mitochondrial dysfunction in tumor cells[59,60]. A key indicator of mitochondrial dysfunction is the reduction in intracellular ATP levels[61]. The result shows that ATP level in CT26 cells is significantly reduced after treatment with RT+MnBTC-Ru, indicating that RT+MnBTC-Ru therapy induced mitochondrial dysfunction (Fig. 4d). Mitochondrial dysfunction leads to a sustained increase in intracellular ROS levels, thereby creating a vicious cycle of mitochondrial damage and ROS accumulation[62]. Additionally, the accumulation of intracellular ROS exacerbates lipid peroxidation (LPO), leading to oxidative damage to cellular membranes. Malondialdehyde (MDA) measurements were used to assess the level of LPO[63], indicating that cell membrane damage was exacerbated following MnBTC-Ru treatment (Fig. 4e). $O_2$ production was indirectly confirmed by evaluating hypoxia-inducible factor 1-alpha (HIF-1α) expression in CT26 cells[64]. Under hypoxic conditions, CT26 cells treated with PBS and MnBTC show a high level of HIF-1α, whereas those treated with MnBTC-Ru exhibit negative HIF-1α signals (Fig. 4f), indicating that the $O_2$ generation by MnBTC-Ru alleviates hypoxia in TME.

Further, we sought to elucidate whether the biocatalytic activity of MnBTC-Ru could enhance DNA damage in the context of RT. The ROS generated by MnBTC-Ru directly induces DNA damage and generates free radicals, while the $O_2$ it produces further reacts with these free radicals, thereby disrupting the DNA repair process[65]. The γ-H2AX, a marker of DNA double-strand breaks, shows a significant increase in cells treated with RT+MnBTC-Ru (Supplementary Fig. 14). This increase may be attributed to membrane damage induced by LPO, which exposes the DNA. Once exposed, the DNA becomes more vulnerable to intracellular ROS and X-ray damage. Additionally, MnBTC-Ru hinders the DNA repair process by producing $O_2$, thereby further leading to permanent DNA damage[66]. These sequential events, from membrane damage to DNA exposure and ultimately to DNA damage, lead to impaired cell function and enhanced cell death. Tumor cell death was evaluated through apoptosis assays and Live/Dead analysis. The RT+MnBTC-Ru group consistently demonstrated the most potent antitumor activity, characterized by the highest apoptosis rate (Fig. 4g

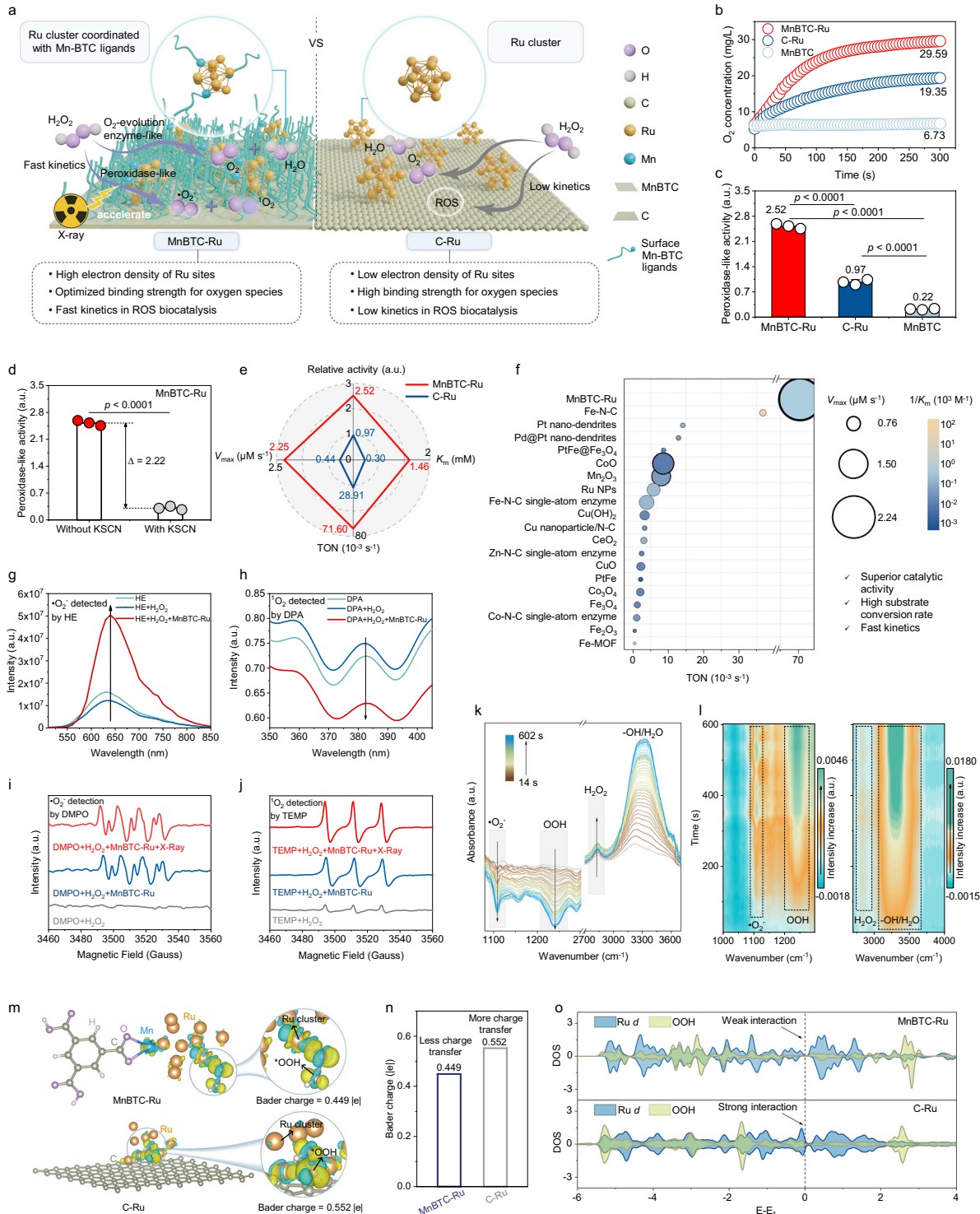

**Fig. 3 | Evaluation of enzyme-mimetic ROS-biocatalytic activities. a** Schematic diagram of the dual-functional capabilities of $O_2$ and ROS generation for MnBTC-Ru, the control biocatalyst (C-Ru) shows sluggish kinetics. **b** The produced $O_2$ concentration that measured by an oxygen dissolving meter with the presence of biocatalysts and $H_2O_2$. **c** Peroxidase-mimetic activity of different catalysts ($n = 3$ independent experiments, data are presented as mean ± SD). **d** Active site poison tests of the biocatalytic MnBTC-Ru complex via KSCN ($n = 3$ independent experiments, data are presented as mean ± SD). **e** Relative peroxidase-like activity, $V_{max}$, $K_m$, and TON values of MnBTC-Ru and C-Ru. **f** Comparison of the TON and $V_{max}$ values with reported enzyme-mimetic catalysts. **g** •$O_2^-$ detected by HE. **h** $^1O_2$ detected by DPA. EPR spectra for recording the **i** •$O_2^-$ signal and the **j** $^1O_2$ signal. **k** In-situ FTIR spectrum and **l** the corresponding contour plot of MnBTC-Ru for the peroxidase-like process. **m** Differential charge density analysis of Ru centers (yellow and cyan represent charge accumulation and depletion, respectively, with a cutoff value of 0.005 e·Bohr$^{-3}$ for the density-difference isosurface). Ru, yellow; Mn, blue; C, khaki; H, white; O, purple. **n** Calculated Bader charge of MnBTC-Ru and C-Ru with the adsorption of a *OOH intermediate. **o** DOS (where Ru corresponds to the $d$ orbital, and OOH is the superposition of $s$ and $p$ orbitals for O and H) of MnBTC-Ru and C-Ru with the adsorption of an *OOH intermediate. Statistical significance was assessed using the Student's t-test for two-group comparisons and one-way ANOVA for multiple-group comparisons, followed by Tukey's two-tailed post-hoc test for pairwise analysis, all tests were two-sided. In (**c–e**, **g–l**), a.u. indicates the arbitrary units. Source data are provided as a Source Data file.

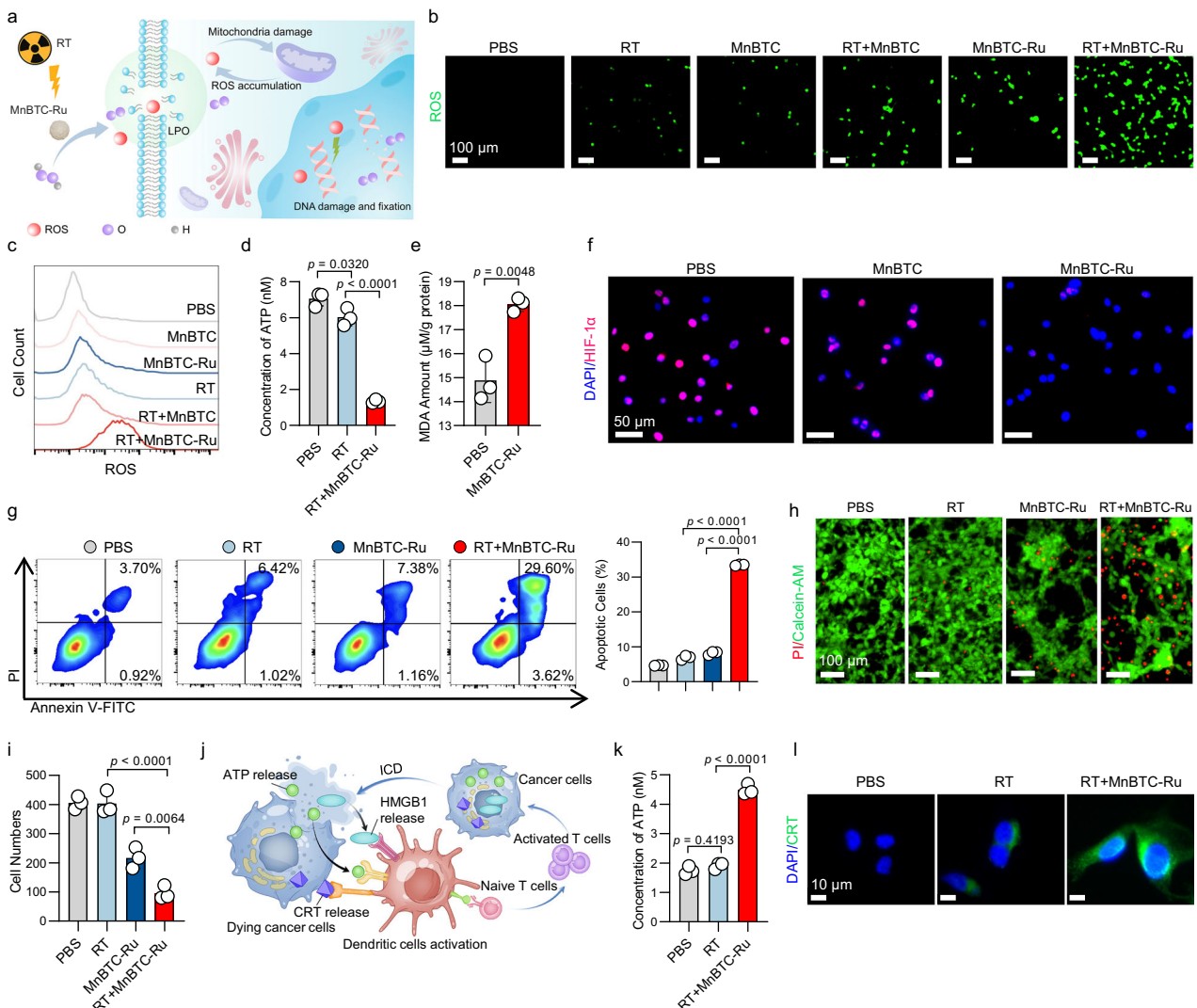

**Fig. 4 | In vitro biocatalytic and radiosensitizing antitumor effects of MnBTC-Ru complex. a** A schematic illustration of the in vitro antitumor activities of MnBTC-Ru. **b** The representative fluorescence images and **c** quantitative flow cytometry show the ROS generation in CT26 cells in different groups ($n = 3$ independent replicates; scale bar = 100 μm). **d** The intracellular ATP of CT26 in different groups ($n = 3$ independent replicates). **e** LPO product MDA detection of CT26 cells after different treatments ($n = 3$ independent replicates). **f** The expression of HIF-1α in CT26 subjected to different treatments (scale bar = 50 μm). **g** The Annexin V/PI analysis of CT26 cells in different groups ($n = 3$ independent replicates). The graph showed the percentage of apoptotic cells (early apoptotic, late apoptotic) in different groups ($n = 3$ independent replicates). **h** Live/Dead analysis of CT26 cells after different treatments, Green: live cells; Red: dead cells (scale bar = 100 μm). **i** Transwell migration assay is used to evaluate cell migration, and the graphs show the numbers of migrated cells in different groups ($n = 3$ independent replicates). **j** A schematic diagram of ICD of tumor cells releasing DAMPs and interacting with dendritic cells. **k** The ATP release of CT26 in different groups ($n = 3$ independent replicates). **l** The representative fluorescence images show the expression of CRT in CT26 after different treatments (scale bar = 10 μm). Experiments were repeated independently (**b, c, f, h, l**) three times with similar results. DAPI indicates 4′,6-diamidino-2-phenylindole, PI indicates propidium iodide, FITC indicates fluorescein isothiocyanate, Calcein-AM indicates calcein acetoxymethyl ester. Results are presented as means ± SD. Statistical significance was assessed using the Student's t-test for two-group comparisons and one-way ANOVA for multiple-group comparisons, followed by Tukey's two-tailed post-hoc test for pairwise analysis, all tests were two-sided. Source data are provided as a Source Data file.

and Supplementary Fig. 15) and intense red fluorescence signals (Fig. 4h). Transwell experiment confirmed the inhibitory effect of MnBTC-Ru on CT26 cell migration, revealing the fewest migrated cells after RT+MnBTC-Ru treatment (Fig. 4i).

After RT+MnBTC-Ru treatment, cells release higher levels of damage-associated molecular patterns (DAMPs), as shown in Fig. 4j[67]. The RT+MnBTC-Ru treatment group exhibits the highest ATP release (Fig. 4k). Additionally, fluorescence imaging shows a significant increase in cell surface calreticulin (CRT) expression after RT+MnBTC-Ru treatment compared to the PBS and RT groups (Fig. 4l). DAMPs are signaling molecules in the immune system, indicating that MnBTC-Ru may enhance systemic antitumor responses and promote clearance of tumor cells[68].

Inspired by the promising in vitro tumoricidal activity of MnBTC-Ru, we further investigate its efficiency in inhibiting primary tumor progression using a BALB/c mouse model bearing CT26 colon carcinoma. The details of the mouse model construction and the treatment strategy are illustrated in Fig. 5a. After post-treatment for 12 days, all mice were sacrificed, and the excised tumors were weighed. While both the MnBTC-Ru and RT treatment individually delay tumor growth compared to the Control group, the combination of RT+MnBTC-Ru exhibits the most significant inhibitory effect on tumor growth (Fig. 5b–d and Supplementary Fig. 16). Throughout the treatment period, the body weight of all groups of mice remains stable (Fig. 5e and Supplementary Fig. 17).

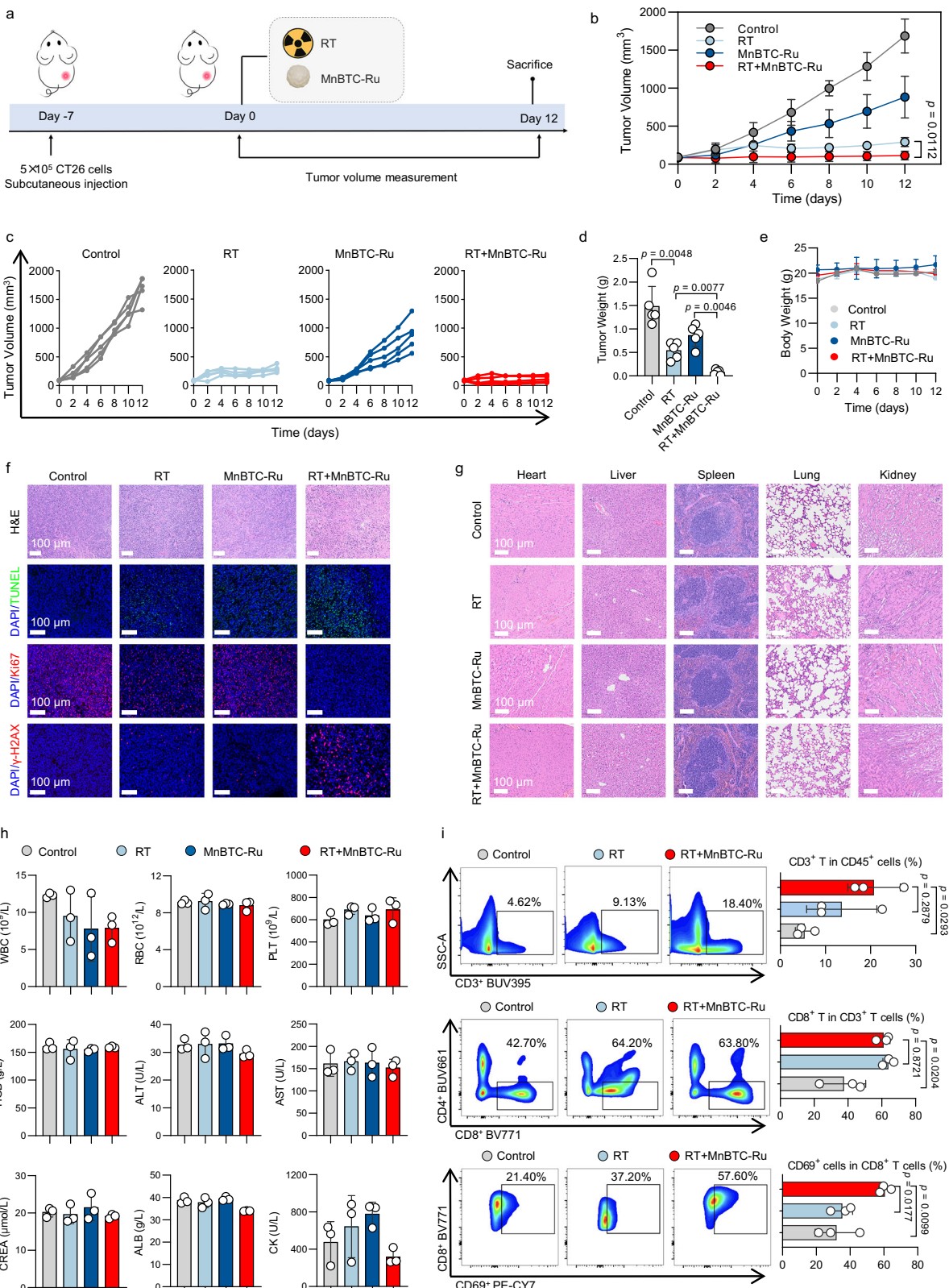

The robust antitumor effect of the RT+MnBTC-Ru treatment is confirmed by histopathologic evaluation (Fig. 5f). The hematoxylin and eosin (H&E) staining results indicate that the tumor tissue exhibits a higher degree of necrosis in the RT+MnBTC-Ru group than in other treatment groups. Additionally, fluorescence images in the terminal deoxynucleotidyl transferase dUTP nick end labeling (TUNEL) assay reveal stronger green fluorescence in the RT+MnBTC-Ru group, indicating enhanced tumor cell apoptosis. Moreover, compared to other treatments, the immunofluorescence results of tumor sections from mice in the RT+MnBTC-Ru treatment group show a significant decrease in Ki67 expression, along with a significant increase in γ-H2AX expression. These findings suggest that RT+MnBTC-Ru treatment causes severe DNA damage, substantially inhibiting tumor cell proliferation.

**Fig. 5 | In vivo tumoricidal effects and enhanced antitumor responses of RT + MnBTC-Ru to inhibit primary tumor progression. a** Schematic illustration of RT +MnBTC-Ru treatment. **b** Average tumor growth curves, **c** individual tumor growth kinetics, **d** tumor weight, and **e** body weight of CT26 tumor-bearing mice after different treatments ($n = 5$ biologically independent mice per group). **f** Representative H&E and fluorescent staining images (TUNEL assay, γ-H2AX and Ki67 staining) from tumor tissue section (scale bar = 100 μm). **g** Representative H&E-stained images from major organ tissue section (scale bar = 100 μm). **h** Results of blood chemistry parameters from mice after different treatments ($n = 3$ independent replicates). Blood chemistry parameters include white blood cell (WBC), red blood cell (RBC), platelet count (PLT), hemoglobin (HGB), alanine aminotransferase (ALT), aspartate aminotransferase (AST), creatinine (CREA), albumin (ALB), and Creatine Kinase (CK). **i** Representative flow cytometric analysis and relative quantification of CD3$^+$ T cells, CD8$^+$ T cells, and CD8$^+$CD69$^+$ T cells ($n = 3$ independent replicates). In (**b**–**i**), the Control group indicates saline. Experiments were repeated independently (**f**, **g**) three times with similar results. Ki67 indicates Marker of Proliferation Ki67, γ-H2AX indicates phosphorylated histone H2A.X at Ser139, SSC-A indicates Side Scatter Area, BUV395 indicates Brilliant Ultraviolet 395, BUV661 indicates Brilliant Ultraviolet 661, BV771 indicates Brilliant Violet 771, PE-Cy7 indicates Phycoerythrin–Cyanine 7. Results are presented as means ± SD. Statistical significance was determined using the one-way ANOVA for multiple-group comparisons, followed by Tukey's two-tailed post-hoc test for pairwise analysis, all tests were two-sided. Source data are provided as a Source Data file.

The safety of RT+MnBTC-Ru treatment was evaluated through H&E staining of major organs and blood chemistry analysis. After treatment, H&E staining was performed on the heart, liver, spleen, lungs, and kidneys of the mice. The results show intact tissue structures and normal cellular morphology, with no evident pathological changes (Fig. 5g and Supplementary Fig. 18). Additionally, blood chemistry analysis confirmed that all measured parameters remained within the normal range, indicating no adverse effects on organ function (Fig. 5h and Supplementary Fig. 19). These results collectively demonstrate the favorable safety profile of RT+MnBTC-Ru treatment.

We further explored the impact of RT+MnBTC-Ru on the antitumor immune system. Flow cytometry was used to determine the proportions of CD3$^+$ T cells, CD8$^+$ T cells, and CD8$^+$CD69$^+$ T cells in tumor tissues (Fig. 5i). Although there are no significant differences in the proportions of CD3$^+$ T cells and CD8$^+$ T cells among the groups, the proportion of CD8$^+$CD69$^+$ T cells significantly increase in tumor tissues treated with RT+MnBTC-Ru. CD8$^+$CD69$^+$ T cells are activated CD8$^+$ cells, suggesting that RT+MnBTC-Ru may enhance the cytotoxic activity of CD8$^+$ cells to eliminate tumors[69]. Previous studies have distinguished three distinct phenotypes within the TME based on the spatial distribution of CD8$^+$ T cells: inflamed, excluded, and desert phenotypes[70,71]. These phenotypes significantly influence the efficacy of immune checkpoint inhibitor therapy. Following the RT+MnBTC-Ru treatment, there is a marked increase in activated CD8$^+$ T cells, transforming the suppressive TME into an inflamed one. These inflamed TME show a favorable response to immune checkpoint inhibitors, particularly anti-PD-L1 and anti-PD-1 therapies[72]. This transformation provides a solid foundation for subsequent anti-PD-1 combination treatments in this study.

After confirming the potent antitumor efficacy and favorable safety profile of MnBTC-Ru, we further investigated its in vivo fate to elucidate the basis of its tumoricidal effects while maintaining minimal systemic toxicity. Quantitative analysis of Ru levels in tumors, major organs, and metabolic excreta was performed using inductively coupled plasma-mass spectrometry (ICP-MS) (Fig. 6a and Supplementary Fig. 20). The results show that at all time points, the percentage of injected dose (%ID) of Ru in tumor tissue remained the highest, significantly exceeding that in other tissues, providing crucial support for its potent radiosensitizing and antitumor effects. In major organs, higher Ru accumulation was observed in the livers and kidneys, suggesting hepatic and renal involvement in MnBTC-Ru metabolism. ICP-MS analysis of urine and feces further confirmed that MnBTC-Ru is primarily excreted via hepatic and renal pathways. Notably, minimal Ru accumulation was detected in the heart, spleen, lungs, and blood, reinforcing the systemic safety of MnBTC-Ru. Furthermore, the reconstruction of CT imaging provides a spatial visualization of MnBTC-Ru in relation to the subcutaneous tumor (Fig. 6b).

To gain deeper insights into the molecular mechanisms underlying RT+MnBTC-Ru therapy for tumor treatment, we perform transcriptome analysis on tumor tissues from three groups: Control, RT, and RT+MnBTC-Ru. Our findings reveal significant variations in gene expression patterns among these groups (Fig. 6c). Compared to the Control group, the RT group exhibits 586 upregulated and 101 downregulated genes, while the RT+MnBTC-Ru group has 1096 upregulated and 321 downregulated genes (Fig. 6d and Supplementary Fig. 21). The top 50 genes are listed in Fig. 6e. To understand MnBTC-Ru's role in the treatment strategy, we specifically identify 862 differentially expressed genes unique to the RT+MnBTC-Ru group for further analysis. Protein-protein interaction (PPI) network analysis of immune-related differentially expressed genes (Supplementary Fig. 22) shows that the RT+MnBTC-Ru combination treatment promotes the formation of a tightly connected network centered on key immune regulators, including IFNG (encoding interferon-γ), TNF (encoding tumor necrosis factor), IL6 (encoding interleukin-6), and PTPRC (encoding CD45 antigen). The strong interactions among these core molecules suggest that the combination therapy reshapes the tumor immune microenvironment by coordinating multiple immune pathways at the molecular level. To further investigate the immunomodulatory effects of RT +MnBTC-Ru, we estimate the composition of immune cell populations within the TME. The RT+MnBTC-Ru group shows a significantly higher immune infiltration score compared to both the Control and RT groups (Supplementary Fig. 23). This increase was primarily driven by a rise in antitumor immune effector cells, accompanied by a reduction in immunosuppressive populations such as M2 macrophages (Supplementary Fig. 24). Such a favorable shift in the immune cell landscape may help explain the improved therapeutic outcomes observed with the RT+MnBTC-Ru treatment. Consistent with these findings, Gene Ontology (GO) functional enrichment analysis reveals that these differentially expressed genes were significantly enriched in immune immune–related biological processes (Fig. 6f). In addition, pathway enrichment analysis based on the Kyoto Encyclopedia of Genes and Genomes (KEGG) database revealed pronounced activation of signaling pathways in the RT+MnBTC combination treatment group, predominantly associated with redox regulation, immune modulation, and inflammation-apoptosis signaling (Supplementary Fig. 25). Moreover, the gene set enrichment analysis (GSEA) result shows activated pathways, such as immune response and immune system process, underscoring the pivotal role of RT+MnBTC-Ru in the activation of antitumor immunity (Fig. 6g). These findings suggest potential synergistic effects between RT+MnBTC-Ru and immune checkpoint inhibitors, presenting an efficient therapeutic avenue for cancer radiosensitization treatment.

**Synergizing immune checkpoint inhibitor against metastases**

Our above findings indicate that RT+MnBTC-Ru effectively stimulates CD8$^+$ T cells within the TME, and the systemic antitumor effects of RT +MnBTC-Ru are further confirmed by RNA sequencing. Consequently, RT+MnBTC-Ru presents great potential as a promising therapeutic strategy that can synergize with anti-PD-1 to enhance antitumor responses, strengthen abscopal antitumor effects, and contribute to

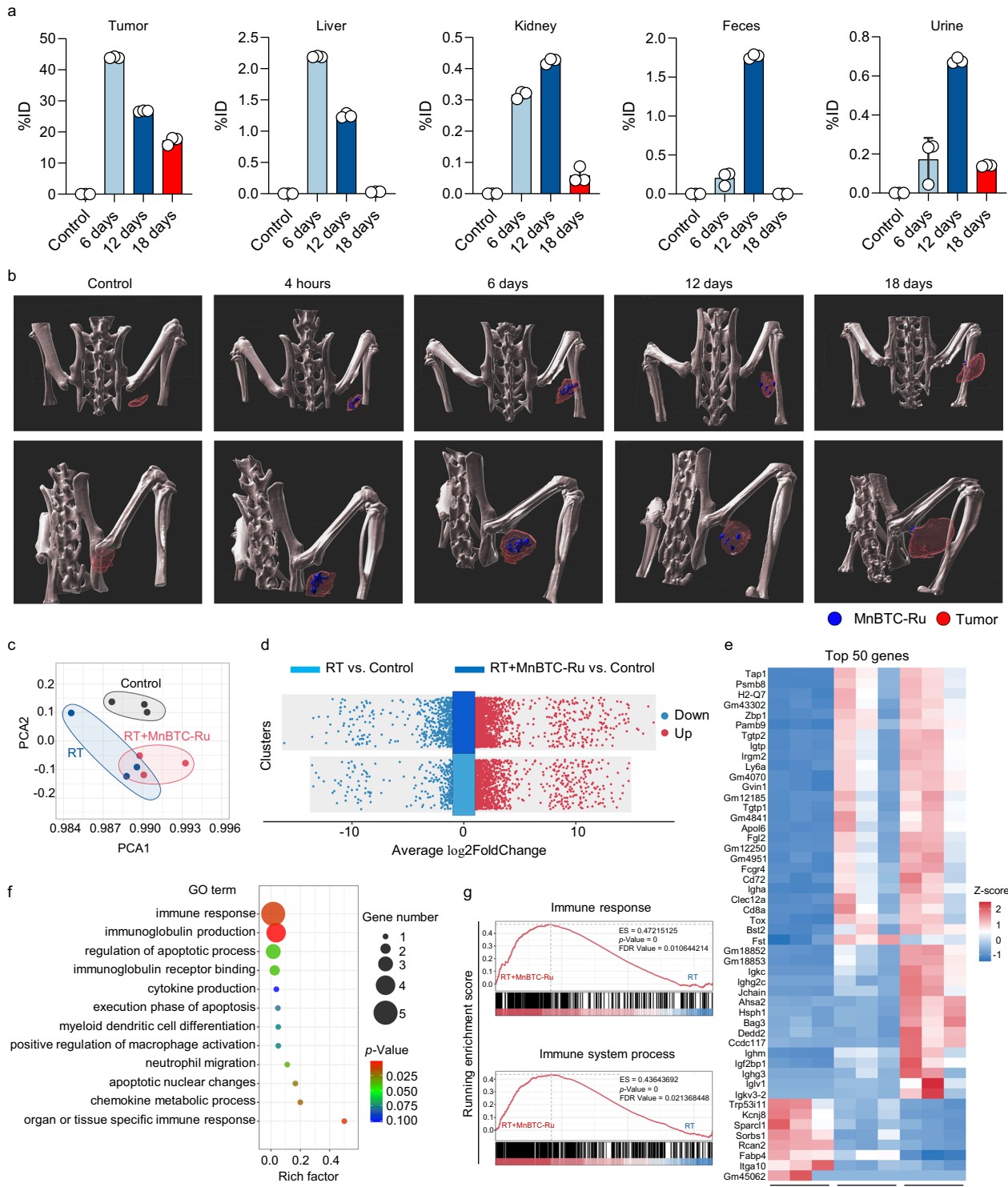

**Fig. 6 | In vivo fate of MnBTC-Ru and its reprogramming of the TME.**
**a** Biodistribution of MnBTC-Ru in the tumor, liver, kidneys, feces, and urine of mice at different time points after intratumoral injection of MnBTC-Ru ($n = 3$ independent replicates). **b** Reconstruction of the skeletal structure, MnBTC-Ru (blue), and tumor (red) in mice based on CT scans at different time points. **c** PCA and **d** Venn diagram of differential expression genes in RNAseq between Control, RT, and RT +MnBTC-Ru treated tumors after 12 days post-treatment ($n = 3$ independent replicates). **e** The top 50 differential expression genes in Control, RT, and RT+MnBTC-Ru groups. Colour gradients have no units; the magnitude of the value indicates the Gene relative expression level. **f** GO enrichment analysis of differentially expressed

genes ($n = 3$ independent replicates). **g** GSEA for the altered gene sets in the RT +MnBTC-Ru treatment group. In (**a-e**), the Control group indicates saline. Experiments were repeated independently (**b**) three times with similar results. PCA indicates principal component analysis, ES indicates enrichment score, FDR indicates false discovery rate. Results are presented as means ± SD. In (**f**), p-value obtained from one-sided Hypergeometric test without multiple comparisons. In (**g**), p-value obtained from two-sided, rank-based permutation test, with significance determined by FDR adjustment for multiple comparisons. Source data are provided as a Source Data file.

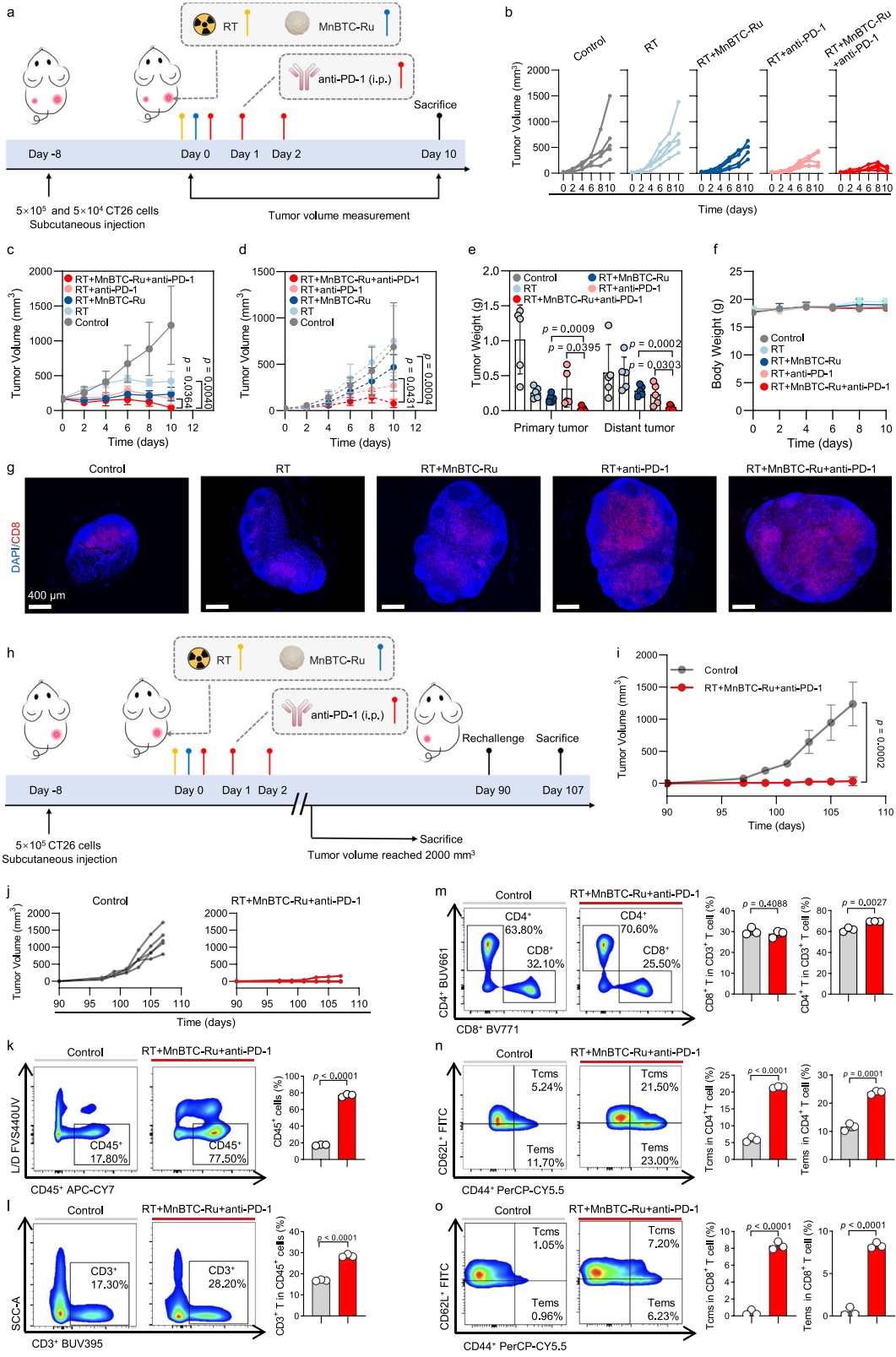

long-term memory effects. Activating systemic antitumor response is crucial in preventing tumor recurrence and metastasis, with the abscopal effect serving as a key indicator[73,74]. Given these observations, we investigate whether RT+MnBTC-Ru can enhance the systemic antitumor response of anti-PD-1.

The establishment of a bilateral tumor-bearing mouse model and the specific administration methods of MnBTC-Ru, RT, and anti-PD-1

are illustrated in the schematic diagram (Fig. 7a). On the 10th day post-treatment, mice were sacrificed, and tumors were excised for weighing (Fig. 7b–e). The growth curve of primary tumors suggests that RT alone can delay tumor progression (Fig. 7c). When RT is combined with either anti-PD-1 or MnBTC-Ru, there is notably enhanced inhibition of tumor growth. The combined treatment of RT+MnBTC-Ru with anti-PD-1 demonstrates the highest efficacy against primary tumors,

**Fig. 7 | RT+MnBTC-Ru and anti-PD-1 combined therapies with abscopal responses and long-term antitumor memory effects to eradicate metastatic tumors. a** Schematic illustration of RT+MnBTC-Ru with anti-PD-1 treatment (i.p., intraperitoneal). **b** Individual tumor growth kinetics of distant tumor, average tumor growth curves of **c** primary tumor and **d** distant tumor, **e** tumor weight, and **f** body weight of bilateral CT26 tumor-bearing mice after different treatments ($n = 5$ biologically independent mice per group). **g** Representative immunofluorescence images from lymph node slices stained with DAPI (blue) and CD8 (red) antibodies (scale bar = 400 μm). **h** Schematic illustration of the experiment design to assess the antitumor memory responses triggered by RT+MnBTC-Ru+anti-PD-1 combination therapy. **i** Average tumor growth curves and **j** individual tumor growth kinetics of the treated mice ($n = 5$ biologically independent mice per group).

**k–o** Representative flow cytometric analysis of Control and RT+MnBTC-Ru+anti-PD-1 group, with corresponding quantification of $CD45^+$, $CD3^+$ T, $CD8^+$ T and $CD4^+$ T cells, and subsets Tcms ($CD62L^+CD44^+$) and Tems ($CD62L^-CD44^+$) from $CD8^+$ and $CD4^+$ T cells in the spleen ($n = 3$ independent replicates). In (**b-g**) and (**i-o**), the Control group indicates saline. Experiments were repeated independently (**g**) three times with similar results. L/D indicates Live/Dead, FVS440UV indicates Fixable Viability Stain 440UV, APC-Cy7 indicates Allophycocyanin–Cyanine 7, PerCP-Cy5.5 indicates Peridinin–Chlorophyll–Protein–Cyanine 5.5. Results are presented as means ± SD. Statistical significance was determined using the Student's t-test for two-group comparisons, and one-way ANOVA for multiple-group comparisons, followed by Tukey's two-tailed post-hoc test for pairwise analysis, all tests were two-sided. Source data are provided as a Source Data file.

leading to complete remission in certain cases. In the growth curve analysis of distant tumors (Fig. 7d), RT alone does not significantly inhibit tumor growth, suggesting a localized effect of RT. As a systemic treatment, anti-PD-1 exhibits a partial inhibitory effect on distant tumors when combined with RT. Incorporating MnBTC-Ru into the treatment regimen produces a synergistic effect, significantly enhancing therapeutic outcomes. The combined therapies of RT+MnBTC-Ru with anti-PD-1 demonstrate the most potent treatment efficacy, offering a more practical approach for tumor treatment compared to other groups. Throughout the treatment, mice in all groups maintained stable body weights (Fig. 7f), and histological examination of major organs stained with H&E show no significant abnormalities, indicating the treatment's safety (Supplementary Fig. 26).

Fluorescent staining images reveals that treatment with RT+MnBTC-Ru combined with anti-PD-1 increases cellular apoptosis and decreases cellular proliferation in tumor tissue (Supplementary Fig. 27). Meanwhile, H&E histological staining shows severe necrosis in tumor slices following this combination therapy, indicating a favorable treatment response. Additionally, immunofluorescence images of pathological sections from draining lymph nodes of distant tumors confirms that the combined treatment of RT+MnBTC-Ru and anti-PD-1 significantly enhances the proliferation of $CD8^+$ T cells within lymph nodes. This combined treatment leads to highly efficient tumor destruction, demonstrating that the combined treatment approach effectively activates the systemic antitumor system, resulting in enhanced tumoricidal effects throughout the body (Fig. 7g).

Specific antitumor memory is a crucial indicator of successful activation of immune system[75]. Effector memory T cells (Tems) and central memory T cells (Tcms) are primary effector cells, enabling them to swiftly and consistently respond to the same antigen upon re-exposure[76]. In our study, BALB/c mice bearing CT26 subcutaneous tumors on the right flank receive RT+MnBTC-Ru+anti-PD-1 treatment. Mice exhibiting tumor regression are monitored, and no tumor recurrence can be detected throughout the observation period. After the 90th day post-treatment, mice are subjected to a tumor rechallenge by implantation of $5 \times 10^5$ CT26 tumor cells into the left flank. Simultaneously, age- and gender-matched untreated mice are also subjected to the same tumor rechallenge, serving as the Control (Fig. 7h). As anticipated, all mice in the Control group display rapid tumor growth upon rechallenge. Conversely, in the combination therapy group, no substantial tumor growth is observed (Fig. 7i, j). Following this, we perform flow cytometry analysis on splenocytes obtained from the mice to examine alterations in the T cell composition. Initially, a notable increase is observed in the proportions of $CD45^+$ cells, $CD3^+$ T cells, $CD3^+CD8^+$ T cells, and $CD3^+CD4^+$ T cells within the group of combination therapy, thus indicating thorough activation of the immune system and underscoring the pivotal role of combination therapy in triggering systemic antitumor responses (Fig. 7k-m). Further analysis on the proportions of Tcms and Tems within $CD4^+$ and $CD8^+$ T cells reveals a significant increase in Tems and Tcms proportions, regardless of whether it is in $CD4^+$ or $CD8^+$ T cells (Fig. 7n, o). This

indicates that even after the cessation of treatment, the antitumor system retains its capability to recognize and combat metastatic tumor cells, thereby bolstering long-term tumor control and preventing recurrence.

## Discussion

In summary, we propose the de novo and bioinspired design of MnBTC-Ru that contains catalytic sites formed by Ru clusters coordinated within the Mn-organic ligands for biocatalytic and radio-sensitization therapies to eradicate primary and metastatic tumors. Our experimental and theoretical investigations reveal that the Mn-organic ligands play a critical role in increasing the electron density of Ru clusters, thereby optimizing the binding affinity of oxygen species and resulting in high capacities for ROS and $O_2$ generation. Accordingly, this biocompatible and biodegradable metal-organic complex enhances RT sensitivity and triggers a transformative response in the TME, thus resulting in a robust antitumor effect. Following RT, MnBTC-Ru induces potent immunogenic cell death (ICD) in tumor cells, releasing DAMPs and signaling effector cells to target the tumor region. Notably, when combined with anti-PD-1 therapy, MnBTC-Ru triggers a strong systemic antitumor response, effectively suppressing both primary and distant tumors and preventing tumor recurrence.

RT remains a cornerstone in the treatment of solid tumors, often in combination with chemotherapy or immunotherapy[77]. However, both intrinsic and acquired resistance to radiation limits its clinical efficacy[78]. Tumor radioresistance arises from multiple factors, including enhanced DNA repair mechanisms in cancer cells[79,80], persistent hypoxia in the TME[81], and the establishment of an immunosuppressive milieu[82]. Together, these factors reduce the responsiveness of tumors to conventional RT, underscoring the need for more effective radio-sensitization strategies. Despite extensive efforts, traditional radio-sensitizers such as metronidazole, misonidazole, and etanidazole have shown limited clinical success when combined with RT[83,84]. These agents lack tumor selectivity and are associated with systemic toxicity. Moreover, the currently developed radiosensitizing materials (e.g., lanthanides, hafnium, and gold) are restricted in future clinical translation due to their insufficient tumoricidal effects and low biodegradability-induced chronic toxicities[14,85].

Our proof-of-concept design paves the way for the development of radiosensitizers that can synergize biocatalytic ROS production with immune system activation, transforming RT into an effective "adjuvant" that enhances antitumor responses, thereby expanding the scope of patients who can benefit from both RT and immune checkpoint inhibitor therapy. The TME is crucial in systemic antitumor responses, with various cells playing key roles in this complex ecosystem[86]. Combining MnBTC-Ru with RT enhances the recruitment of dendritic cells by providing more DAMPs through ICD, improving the therapeutic outcomes of RT[87]. Here, we have developed MnBTC-Ru as a TME-adaptive and biocatalytic radiosensitizer for enhancing radiosensitivity, modulating the TME, and enhancing tumor antigenicity.

The integration of ROS biocatalysis, $O_2$ sensitization, and bioinspired metal-organic coordination-based radiosensitizers presents a promising avenue for advancing biocatalytic and radiotherapeutic strategies in malignant tumors. This approach holds significant potential, particularly in preventing tumor metastasis and recurrence. To realize its full clinical potential, future research should prioritize systematically evaluating the long-term biocompatibility and biodegradability of these biocatalytic and radioactive materials. Additionally, a deeper understanding of their mechanisms of action within complex biological environments is essential to optimize their therapeutic application. These efforts will not only provide a robust scientific basis for uncovering the full scope of their therapeutic capabilities but also accelerate their effective translation into clinical use. By bridging basic research and clinical implementation, this direction has the potential to revolutionize cancer treatment, offering patients more precise and effective therapeutic solutions.

## Methods

### Materials
The ruthenium chloride hydrate ($RuCl_3 \cdot xH_2O$) and o-phenylenediamine (OPD) were obtained from Energy Chemical (Shanghai, China). Manganese (II) nitrate tetrahydrate ($Mn(NO_3)_2 \cdot 4H_2O$), 1,3,5-benzene-tricarboxylic acid (BTC), sodium borohydride ($NaBH_4$), hydroethidine (HE), and 9,10-diphenanthraquinone (DPA) were obtained from Aladdin reagents (Shanghai, China). All other chemicals were obtained from Aladdin reagents and used as received without further purification.

### Synthesis of MnBTC-Ru
To obtain MnBTC, $Mn(NO_3)_2 \cdot 6H_2O$ (192 mg) and BTC (87.5 mg) were individually added to 25 mL of ethanol, followed by 30 min of stirring. The combined solution was then transferred to a 100 mL Teflon-lined autoclave. After stirring for 0.5 h, the mixture was sealed and heated at 160 °C for 12 h. The resulting precipitates were centrifuged, washed thrice with ethanol, and dried at 60 °C under vacuum for 12 h. To obtain MnBTC-Ru, 50 mg of MnBTC was re-dispersed in 25 mL of ethanol using ultrasonic treatment for 20 min. Subsequently, 0.5 mL of a $RuCl_3 \cdot 3H_2O$ aqueous solution (10 mg mL$^{-1}$) was added dropwise to the homogeneous precursor solution and stirred at 30 °C for 12 h. The mixture was washed with ethanol, dried overnight in a vacuum oven at 60 °C, and labeled as MnBTC-Ru.

### Synthesis of C-Ru
$Mn(NO_3)_2 \cdot 6H_2O$ (2.4 mg), $RuCl_3 \cdot 3H_2O$ (20 mg), carbon black (200 mg), and $NaBH_4$ (36.5 mg) were individually added to 25 mL of deionized pure water, followed by 30 min of stirring. The mixture was washed with ethanol, dried overnight in a vacuum oven at 60 °C, and labeled as C-Ru.

### Dynamic light scattering test
10 μg of biocatalysts were dissolved in 1 mL of either PBS or mouse serum, and dynamic light scattering analysis was conducted utilizing the Malvern Nano-ZS instrument.

### Peroxidase-mimicking activity measured by TMB
The biocatalysts solution (10 mg·mL$^{-1}$, 10 μL) was added into NaOAc−HOAc buffer (100 mM, pH 4.5) containing $H_2O_2$ (0.1 M, 25 μL) and TMB (10 mg·mL$^{-1}$, 24 μL). The final volume of the mixture was adjusted to 2 mL with NaOAc−HOAc buffer; the final sample concentration was 50 μg/mL. Then, 200 μL of the mixture was used for UV−visible spectroscopy measurements at an absorbance of 652 nm.

### Enzyme dynamic parameters
The Michaelis-Menten constant was calculated based on the Michaelis-Menten saturation curve. For each concentration of $H_2O_2$, the initial reaction rates ($v$) of the oxidation of TMB were calculated from the

variation in absorbance using the Beer-Lambert Law (Eq. (1)), which has an $\varepsilon$ of 39,000 M$^{-1}$ cm$^{-1}$ for oxidized-TMB (ox-TMB) and a $b$ value of 1 cm for the length of the solution in the light path, where $c$ indicates the ox-TMB concentration and $A$ represents the absorbance. The reaction rates were then plotted against their corresponding $H_2O_2$ concentration and then fitted with the Michaelis-Menten curves (Eq. (2)). Furthermore, a linear double-reciprocal plot (Lineweaver-Burk plot, Eq. (3)) was used to determine the maximum reaction velocity ($V_{max}$) and Michaelis constant ($K_m$). Furthermore, the turnover number (TON) was calculated according to Eq. (4).

$$c = \frac{A}{\varepsilon b} \tag{1}$$

$$v = \frac{V_{max} \cdot [S]}{K_m + [S]} \tag{2}$$

$$\frac{1}{v} = \frac{K_m}{V_{max}} \cdot \frac{1}{[S]} + \frac{1}{V_{max}} \tag{3}$$

$$TON = \frac{V_{max}}{[E_0]} \tag{4}$$

[S] is the concentration of $H_2O_2$, and [$E_0$] is the molar concentration of metal in nanozymes.

### Active site poison tests by KSCN
30 mg of KSCN were dissolved in water, followed by the addition of 10 μL of a biocatalyst solution at a concentration of 10 mg/mL. After a duration of 1 h, the mixture underwent centrifugation, subsequent to which the standard peroxidase-like activity assay was conducted.

### Analysis of free radicals by quenching experiments
To investigate •OH, a mixture was prepared by adding tert-butanol (TBA, 200 μL), $H_2O_2$ (25 μL, 0.1 M), catalysts (10 μL, 10 mg mL$^{-1}$), and TMB (24 μL, 10 mg mL$^{-1}$) to NaOAc/HOAc buffer (1740 μL, 100 mM, pH 4.5). After a 10 min reaction period, the absorbance at 652 nm was measured.

To investigate •$O_2^-$, a mixture was prepared by adding p-benzoquinone (BQ, 100 μL), $H_2O_2$ (25 μL, 0.1 M), catalysts (10 μL, 10 mg mL$^{-1}$), and TMB (24 μL, 10 mg mL$^{-1}$) to NaOAc/HOAc buffer (1840 μL, 100 mM, pH 4.5). After a 10 min reaction period, the absorbance at 652 nm was measured.

To investigate $^1O_2$, a mixture was prepared by adding sodium azide ($NaN_3$, 100 μL), $H_2O_2$ (25 μL, 0.1 M), catalysts (10 μL, 10 mg mL$^{-1}$), and TMB (24 μL, 10 mg mL$^{-1}$) to NaOAc/HOAc buffer (1840 μL, 100 mM, pH 4.5). After a 10 min reaction period, the absorbance at 652 nm was measured.

### Detection of •$O_2^-$ by HE
The HE probe was utilized to measure the levels of •$O_2^-$ in a system, which could react with •$O_2^-$ to produce fluorescent Ethidium at the wavelength of 470 nm and an emission wavelength of 610 nm. The experimental procedure involved mixing 1.5 mL of a catalyst solution (100 μg/mL) with 1.5 μL of 0.1 M $H_2O_2$ at 37 °C for 40 min. Subsequently, 1.5 mL of a HE-ethanol solution (1 mg/mL) was added to the system and thoroughly vortexed. The solution was allowed to sit undisturbed for 40 min before being subjected to fluorescence measurements using a Synergy Mx device.

### Detection of $^1O_2$ by DPA
In the detection of $^1O_2$, a mixture was prepared by adding 25 μL of a 4 mg/mL catalyst solution, 25 μL of a 0.1 M $H_2O_2$ solution, and 100 μL of a DPA-Dimethyl sulfoxide (DMSO) solution into a 2 mL DMSO

solution. The mixture was then analyzed using a UV-vis spectrophotometer.

## EPR Measurement

The generation of $^1O_2$ was assessed using 2,2,6,6-Tetra-methylpiperidine 1-oxyl (TEMP). A mixture was prepared by adding catalysts (10 μL, 10 mg mL$^{-1}$) and $H_2O_2$ (10 μL, 10 M) to NaOAc/HOAc buffer (500 μL, 100 mM, pH 4.5), followed by the addition of 20 μL of TEMP.

The generation of $\cdot O_2^-$ was assessed using 5,5-dimethyl-1-pyrro-line N-oxide (DMPO). A mixture was prepared by adding catalysts (10 μL, 10 mg mL$^{-1}$) and $H_2O_2$ (10 μL, 10 M) to DMSO (500 μL), followed by the addition of 10 μL of DMPO.

## $O_2$ generation capability

The capability for $O_2$ generation was assessed using the following procedure: biocatalysts (20 μL, 10 mg mL$^{-1}$) and $H_2O_2$ (200 μL, 10 M) was added to PBS (20 mL, pH 6.5). The concentration of dissolved $O_2$ was measured over a 3 min period using a dissolved oxygen meter, with readings taken at 5 s intervals.

## X-ray-activated enzyme-like activity

10 mg of OPD was dissolved in 1 mL of N,N-Dimethylformamide. Then, $H_2O_2$ (10 μL, 0.1 M), OPD (24 μL, 10 mg/mL), and different materials (20 μL, 4 mg/mL) were successively added to 1945 μL of NaOAc/HOAc buffer (100 mM, pH 5.5). 200 μL of the resulting solution was transferred into a 96-well plate. After irradiation with X-rays at 6 Gray (Gy), the absorbance was measured at the maximum absorption wavelength of 448 nm.

## Cell lines and animals

The CT26 cell line (Cat CRL-2639) was sourced from the American Type Culture Collection (ATCC) and verified for authenticity through short tandem repeat analysis. CT26 cells were cultured in RPMI 1640 medium (Gibco) containing 1% penicillin (Boster), 1% streptomycin (Boster), and 10% fetal bovine serum (FBS, VivaCell) at 37 °C in 5% $CO_2$ humidified air. For subsequent experiments, cells were immediately placed in an anaerobic incubator (Thermo Scientific, USA) filled with 94.5% $N_2$, 5% $CO_2$, and 0.5% $O_2$ after seeding to induce hypoxia. Female BALB/c mice (6–8 weeks old) were purchased from Jiangsu Gem-pharmatech Co., Ltd. and housed in a specific pathogen-free facility. All mice were maintained under controlled conditions with a 12 h light/dark cycle (lights on from 8:00 a.m. to 8:00 p.m.), constant temperature (18 °C–22 °C), and humidity (50–60%), with unrestricted access to standard chow and water. All diets were supplied by Jiangsu Xietong Inc. (Nanjing, China). The animal experiments were approved by the Institutional Animal Care and Use Committee of West China Hospital, Sichuan University (Approval No. 20241017001). All animal experiments were conducted in accordance with the ARRIVE guidelines.

## Intracellular ROS detection

The production of ROS was determined by the ROS probe DCFH-DA (Beyotime, China). CT26 cells were pretreated with PBS, MnBTC, or MnBTC-Ru at an equivalent manganese concentration of 4 μg/mL for 4 h followed by irradiation with X-ray at 6 Gy (RS2000, Rad Source, USA; 160 kV, 25 mA, 1 mm Cu filter). After different treatments, cells were treated with DCFH-DA (10 μM) for 30 min before imaging by the Olympus IX83 Live Microscope (Olympus Life Science, Japan). Additionally, cells subjected to different treatments were collected and incubated with DCFH-DA (10 μM) for quantitative ROS detection using flow cytometry (CytoFLEX, Beckman, USA).

## ATP detection

The intracellular ATP and released ATP from different treatments were quantified using an enhanced ATP assay kit (Beyotime, China). The fluorescence signal was collected with a BioTek Synergy™ Mx microplate reader.

## MDA detection

The MDA levels were determined using a lipid peroxidation assay kit (Beyotime, China). Protein concentrations of the cell lysates were quantified with a Bicinchoninic Acid protein assay kit (Beyotime, China).The fluorescence signal was collected with a BioTek Synergy™ Mx microplate reader.

## Detection of HIF-1α expression

Immunofluorescence staining was performed to assess HIF-1α expression. CT26 cells were pretreated with PBS, MnBTC, or MnBTC-Ru. Cells were fixed with 4% formaldehyde, permeabilized using Triton X-100, and blocked with 1% Bovine Serum Albumin (BSA). Subsequently, they were incubated overnight at 4 °C with an anti-HIF-1α antibody (Abcam, Catalog No. ab179483, 1:500 dilution), followed by a 1 h incubation with a fluorescent secondary antibody. Nuclei were counterstained with DAPI for 10 min prior to imaging.

## Apoptosis/necrosis assays

CT26 cells were cultured in six-well plates overnight and incubated with PBS, MnBTC, or MnBTC-Ru for 4 h followed by irradiation with X-ray at 6 Gy. After 48 h, the cells were stained according to the Annexin V-FITC/PI cell apoptosis kit (4 A BIOTECH, China) and quantified by flow cytometry (CytoFLEX, Beckman, USA).

## Live/dead assay

CT26 cells were cultured in 24-well plates overnight and incubated with PBS or MnBTC-Ru followed by irradiation with X-ray at 6 Gy. After 48 h, the cells were stained according to the Calcein/PI Live/Dead Viability Assay Kit (Beyotime, China) and imaged by the Olympus IX83 Live Microscope (Olympus Life Science, Japan).

## Cell migration assays

Cell migration was evaluated using 6.5 mm polyethylene ter-ephthalate membrane transwell chambers with 8.0 μm pores (Labselect, China). CT26 cells were pretreated with PBS or MnBTC-Ru at an equivalent manganese concentration of 4 μg/mL for 4 h followed by irradiation with X-ray at 6 Gy. Complete medium containing 10% fetal calf serum was placed in the lower chamber, while serum-free medium was added to the upper chamber. After 24 h of incubation, non-migrated cells on the upper surface were gently removed, and migrated cells on the lower side of the membrane were fixed with 4% formaldehyde and stained using 0.1% crystal violet prior to quantification.

## Detection of CRT expression

The expression of CRT was examined using immunofluorescence assays. After various treatments, cells were fixed with 4% formaldehyde and blocked with 1% BSA. Subsequently, they were incubated overnight at 4 °C with an anti-CRT antibody (Abcam, Catalog No. ab92516, 1:500 dilution), followed by a 1 h incubation with a fluorescent secondary antibody. Nuclei were counterstained with DAPI for 10 min prior to imaging.

## γ-H2AX immunofluorescence assay

CT26 cells were pre-seeded in a 96-well plate overnight and incubated with PBS, MnBTC, or MnBTC-Ru at an equivalent manganese concentration of 4 μg/mL for 4 h followed by irradiation with X-ray at 6 Gy. After irradiation, immunofluorescence staining for γ-H2AX was performed using the DNA damage assay kit for γ-H2AX immunofluorescence (Beyotime, China). Images were acquired using the Olympus IX83 Live Microscope (Olympus Life Science, Japan).

## Tumor models and treatment experiments

To establish a unilateral tumor model, we subcutaneously injected $5 \times 10^5$ CT26 cells into the right flank of female BALB/c mice (6-8 weeks old). Treatment began when the tumor size reached 80–100 mm³ (7 days after inoculation). The mice were randomly divided into six groups: Control, MnBTC, MnBTC-Ru, RT, RT+MnBTC, and RT+MnBTC-Ru. Four hours before RT, mice in each group received intratumoral injections of Control/MnBTC/MnBTC-Ru (equivalent manganese doses of 40 μg per mouse). For groups undergoing RT, mice were anesthetized with 1% pentobarbital sodium (100 μL per 18 g body weight) and placed in individual lead boxes to expose the tumor area to irradiation while shielding the rest of the body. RT was administered using an X-ray irradiator (RS2000, Rad Source, USA) with parameters set at 160 kV and 25 mA. Following treatment, tumor size, and body weight were measured every other day. Tumor volume (mm³) was calculated using the formula (length × width²) × 0.5. Euthanasia was performed if the tumor volume exceeded 2000 mm³ or if signs of distress were observed.

To establish a bilateral tumor model, we subcutaneously injected $5 \times 10^5$ CT26 cells into the right flank of female BALB/c mice as the primary tumor and $5 \times 10^4$ cells into the left flank as the distant tumor. Treatment began when the tumor size reached 150–200 mm³ (8 days after inoculation). The mice were randomly divided into five groups: Control, RT, RT+MnBTC-Ru, RT+anti-PD-1, and RT+MnBTC-Ru+anti-PD-1. Four hours before RT, mice in each group received intratumoral injections of Control/MnBTC-Ru in the primary tumor. In vivo anti-PD-1 antibody was obtained from BeiGene (Ch15mt). For mice in groups receiving anti-PD-1 treatment, anti-PD-1 was administered via intraperitoneal injection (20 mg/kg on Day 0, 10 mg/kg on Day 1 and Day 2). The RT of the primary tumor and the subsequent monitoring of bilateral tumors were the same as the aforementioned procedures.

To establish an immune memory model, mice with unilateral tumors were treated with RT+MnBTC-Ru+anti-PD-1. Mice exhibiting complete tumor regression after this treatment were considered effectively treated and were used for further analysis. The specific tumor inoculation methods and treatment procedures are as described above. Age- and sex-matched naive mice were set up as a Control group. On day 90, a tumor rechallenge was conducted by inoculating $5 \times 10^5$ CT26 tumor cells into the left flank of the effective treatment model and naive mice. The subsequent monitoring of tumors was the same as the aforementioned procedures.

## In vivo fluorescence imaging

Tumor tissues collected from mice were fixed in 4% formalin for 48 h, then embedded in paraffin, and sectioned into 4 μm slices. The slides were subjected to the TUNEL assay following established protocols from previous studies. For immunofluorescence, tissue sections were first treated with anti-γ-H2AX antibody (Servicebio, Catalog No. GB111841, 1:100 dilution), followed by labeling with Cy3-conjugated goat anti-rabbit IgG (Servicebio, Catalog No. GB21303). Another set of tissue sections was treated with anti-Ki67 antibody (Servicebio, Catalog No. GB111141, 1:500 dilution) and labeled with Cy3-conjugated goat anti-rabbit IgG (Servicebio, Catalog No. GB21303). Additionally, tissue sections were treated with anti-CD8 antibody (Servicebio, Catalog No. GB15068, 1:1000 dilution) and labeled with Cy3-conjugated goat anti-rabbit IgG (Servicebio, Catalog No. GB21303). Images were acquired using the Olympus VS200 (Olympus Life Science, Japan).

## In vivo biocompatibility study

To evaluate biocompatibility, blood samples were obtained from mice on day 12 after treatment and subjected to blood chemistry analysis by Lilai Biotechnology Co., Ltd. (Chengdu, China). Simultaneously, major organs, including the heart, liver, spleen, lungs, and kidneys—were harvested, fixed in formalin, embedded in paraffin, sectioned, and stained with H&E following standard procedures. Tissue images were captured using an Olympus VS200 (Olympus Life Science, Japan).

## In vivo biodistribution study

For the biodistribution study, BALB/c mice bearing CT26 subcutaneous tumors were respectively injected with MnBTC-Ru. Computed Tomography (CT) scans (NEMO micro-CT, Pingseng Scientific, China) were performed at five-time points: before injection and at 4 h, 6 days, 12 days, and 18 days post-injection. To visualize the spatial relationship between the MnBTC-Ru and the subcutaneous tumor, DICOM images obtained from CT scans were processed and reconstructed into 3D models using Imaris software (10.2).

Additionally, at 6, 12, and 18 days post-injection, major organs (including the heart, liver, spleen, lungs, and kidneys), tumor tissues, and key metabolic excreta (urine and feces) were collected. Following accurate weighing, each sample was digested in aqua regia at 80 °C until complete dissolution was achieved. The resulting mixture was centrifuged at 12,000×g for 20 min, and the supernatant was collected for subsequent analysis by inductively coupled plasma mass spectrometry (ICP-MS, Agilent 7850).

## Flow cytometry analysis

The tumor tissues and spleen of CT26 tumor-bearing BALB/c mice after different treatments were retrieved for flow cytometry analyses. Single-cell suspensions were prepared from excised tumor tissues or spleen, followed by incubation with FVS440UV (BD, Catalog No. 566332, 1:1000 dilution) for 30 min to assess cell viability. Subsequently, the cell suspension was incubated with purified Rat Anti-Mouse CD16/CD32 (BD, Catalog No. 553141, 0.5 μg per test) to block nonspecific binding, and then stained with anti-CD45-APC-CY7 (BD, Catalog No. 557659), anti-CD3-BUV395 (BD, Catalog No. 563565), anti-CD4-BUV661 (BD, Catalog No. 741461), anti-CD8-BV771 (BD, Catalog No. 563046), anti-CD69-PE-CY7 (BD, Catalog No. 552879), anti-CD44-PerCP-Cy5.5 (BD, Catalog No. 560570), anti-CD62L-FITC (BD, Catalog No. 561917). All antibodies were prepared at a final concentration of 0.2 μg per test. After staining, cell suspensions were filtered and analyzed using a FACS FCM (FACSymphony A5, BD, USA), and data were processed with FlowJo software (version 10.8.1).

## Statistical analysis

Statistical analyses were performed using GraphPad Prism 10.2.3 software (GraphPad Software Inc.). Experimental parameters, including sample size ($n$), statistical significance ($p$), data normalization protocols, and specific statistical tests, are comprehensively detailed in the corresponding images and figure legends. Quantitative data were collected at least three independent times. All data are expressed as the mean values ± SD. Statistical significance was determined using the Student's $t$ test for two-group comparisons, and one-way ANOVA for multiple-group comparisons, followed by Tukey's two-tailed post-hoc test for pairwise analysis, all tests were two-sided.

## Reporting summary

Further information on research design is available in the Nature Portfolio Reporting Summary linked to this article.

# Data availability

All data supporting the results of this study are available within the paper and its Supplementary Information. All raw data generated for the figures in this study are provided in the source data file. Source data are available for Figs. 1c, g, k, 2a–h, 3b–l, n, o, 4d, e, g, i, k, 5b–e, h, i, 6a, 7b–f, i–o, and Supplementary Figs. 1, 4, 8, 9, 10, 11, 12, 13, 15, 16, 17, 19, 20, 21, and 23 in the associated source data file. The raw sequencing data generated in this study have been deposited in the NCBI Sequence Read Archive (SRA) under the BioProject accession number PRJNA1288257. These data are publicly available and can be accessed

through the NCBI SRA database. Source data are provided with this paper.

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

## Acknowledgements

The work was supported by the Noncommunicable Chronic Diseases-National Science and Technology Major Project (2023ZD0503000 [X.C.P.] and 2023ZD0503004 [X.C.P.]), the Regional Innovation and Development Joint Fund Key Project of the National Natural Science Foundation of China (U24A20735 [X.C.P.]). The National Natural Sciences Foundation of China (82473434 [X.C.P.], 52173133 [C.C.], 52373148 [C.C.], 52525311 [C.C.]), Sichuan Provincial Science and

Technology Department Key Research and Development Program (2022YFSY0012 [X.C.P.]), Sichuan Science and Technology Program (2024YFHZ0041 [Q.L.W.], 2024ZYD0054 [X.C.P.]), Science and Technology Project of Sichuan Provincial Health Commission (Clinical Research Special Project JH2023082 [X.C.P.]), the International Science and Technology Cooperation Program of Chengdu Science and Technology Bureau (2024-YF06-00011-HZ [X.C.P.], 2022-GH03-00004-HZ [X.C.P.]), the Health Research Project of Chengdu Eastern New Area Management Committee (202304 [X.C.P.]), 1.3.5 project for disciplines of excellence from West China Hospital of Sichuan University (ZYYC23006 [X.C.P.]), the Ministry of Education University-Industry Collaborative Education Program (230720523707281 [X.C.P.]). The funders had no role in study design, data collection and analysis, decision to publish, or preparation of the manuscript. We sincerely appreciate the valuable assistance and guidance provided by Na Chen from the Department of Biotherapy, Cancer Center, and the State Key Laboratory of Biotherapy at West China Hospital, Sichuan University. We also thank Ting Cao, Yan Wang, Jian Yang, Dan Li, Jiehao Chen, Qiuxiao Shi, Linqiao Tang, Wanli Zhang, Huifang Li, Xiangyi Ren, Mengli Zhu, Li Zhou, Jingyao Zhang and Cong Li from the Core Facilities of West China Hospital for their assistance and guidance. We would like to thank Dr. Hanjiao Chen of the Analytical & Testing Center of Sichuan University for their assistance on EPR. We further acknowledge Zujie Li and Xiaofeng Lin from LC Bio Technology Co., Ltd. for their assistance with the bioinformatics analysis of RNA-seq data.

## Author contributions

R.D.L. and T.W. contributed equally to this work. R.D.L. and T.W. performed the experiments and analyzed the results. Z.Y.X. designed and conducted the theoretical calculation. R.D.L., T.W., S.D.M., Z.Y.D., Q.L.W., Z.G.W., X.L.W., M.A., S.L., X.C.P. and C.C. assisted with the figure production and experiment design. R.D.L., T.W., X.C.P. and C.C. wrote and edited the manuscript. X.C.P. and C.C. designed the experiments and supervised the whole project. X.C.P. and C.C. contributed to the review and editing of the final manuscript. All authors discussed the results and commented on the manuscript.

## Competing interests

The authors declare no competing interests.
