## [Transparent Peer Review file · Nature Communications]

Bioinspired ruthenium-manganese-oxygen complex for biocatalytic and radiosensitization therapies to eradicate primary and metastatic tumors

Corresponding Author: Professor Chong Cheng

Version 1:

Reviewer comments:

Reviewer #1

(Remarks to the Author)

This work developed a bioinspired RuMn-oxygen complex (MnBTC-Ru) for biocatalytic and radiosensitization therapies aimed at eradicating primary and metastatic tumors. Through comprehensive experimental data, theoretical calculations, and in situ FTIR experiments, the authors elucidated the catalytic mechanism of the unique Mn-organic ligands in producing ROS. They also demonstrated that MnBTC-Ru could enhance the sensitivity to X-ray irradiation, increase DNA damage in cancer cells, and, when combined with anti-PD-1 therapy, promote robust systemic antitumor responses while preventing tumor recurrence and metastasis. The study was comprehensive, its concept is new, and its figures were well prepared. Nevertheless, the authors may take into account the following aspects in revising the paper.

- (1) About the simulation of the natural Mn-POD active center in MnBTC-Ru, what advantages did the addition of Ru offer to radiosensitization therapy? How is Ru in comparison with Os or Ir?
- (2) It would be beneficial to include further DLS testing to assess the dispersibility of MnBTC-Ru, as this will help determine any potential adverse effects in vivo.
- (3) While in situ infrared spectroscopy is a valuable characterization method, validating these findings against existing literature would enhance the analysis, some references should be added to validate the peaks.
- (4) The authors have made progress in verifying the mechanism of $^{\bullet}\text{OOH}$ interaction between different material centers; however, further clarification on why $^{\bullet}\text{OOH}$ serves as a key intermediate in forming $^{\bullet}\text{O}_2^-$ and 1O_2 would be helpful.
- (5) Two important issues regarding the biological mechanisms of this system are missing. First, more in-depth analysis is needed on the role of this system in activating antitumor immunity, especially given the gene sequencing data; second, how is the selectivity of this system when acting on tumor cells against normal cells?
- (6) More discussion from a clinical translation perspective is preferred on the current limitations of tumor radiotherapy and how this system could potentially tackle these issues and accelerate clinical application.

Reviewer #2

(Remarks to the Author)

The paper 《Bioinspired Ruthenium-Manganese-Oxygen Complex for Biocatalytic and Radiosensitization Therapies to Eradicate Primary and Metastatic Tumors》 has developed a novel MOF anti-tumor material, which has certain practical significance for tumor treatment. However, there are still some problems, mainly as follows:

1. The characterization results of MnBTC are inconsistent with those reported in this literature, especially XRD. Please explain the reasons. Chen, Y., Tian, Q., Wang, H. Y., Ma, R. N., Han, R. T., Wang, Y., Ge, H. B., Ren, Y. J., Yang, R., Yang, H. M., Chen, Y. J., Duan, X. Z., Zhang, L. B., Gao, J., Gao, L. Z., Yan, X. Y., & Qin, Y. (2022). A Manganese-Based Metal-Organic Framework as a Cold-Adapted Nanozyme. *Advanced Materials*.
2. What is the basis for drawing the structure of Figure 1b? There should be Mn³⁺ (referred to as Mn⁴⁺ in the text) here, otherwise how is ROS generated? This structural diagram should not reflect the oxidation effect of Mn ions.
3. Figure 2b shows the appearance of Mn⁴⁺ in MnBTC, which is somewhat unbelievable. It is rare for Mn²⁺ to directly change to Mn⁴⁺ in MOFs. Please verify that in general, the Mn in MnBTC structure is Mn³⁺, unless MnBTC is synthesized from Mn³⁺ as the raw material. However, this is clearly not the case in this article (it is Mn(NO₃)₂).
4. How to achieve targeted therapy for tumors after introducing Mn ions in this study, and will there be stronger potential side

effects?

5. In this study, there were 5 mice in each treatment group, and the data error was very small. However, the individual differences among mice are generally large. Can such a small error be made (Figure 5b, 5c, 5d)? Especially for the control group in Figure 5c, please provide a detailed explanation of the experimental and computational processes.

6. This article conducted transcriptomic analysis, but why is there no detailed KEGG analysis? KEGG analysis can provide a clearer explanation for the cellular pathway changes caused by MnBTC and MnBTC-Ru.

7. The proportion of ROS produced by each component should be calculated, indicating which component plays a key role and through which signaling pathway it acts on cells.

Version 2:

Reviewer comments:

Reviewer #1

(Remarks to the Author)

The authors have adequately addressed my questions with a careful revision, accompanied by new data and thorough discussion. I now recommend a favourable consideration of the manuscript.

Reviewer #2

(Remarks to the Author)

The author has revised and resolved the vast majority of issues, and the current status of the article has reached the publishing level.

In addition, please carefully check the continuity and completeness of the modified and unmodified parts.

Point-by-point response to the detailed comments by reviewers of “*Bioinspired Ruthenium-Manganese-Oxygen Complex for Biocatalytic and Radiosensitization Therapies to Eradicate Primary and Metastatic Tumors*” with manuscript ID: NCOMMS-25-08214A-Z.

REVIEWER COMMENTS

Reviewer #1 (Remarks to the Author):

“This work developed a bioinspired RuMn-oxygen complex (MnBTC-Ru) for biocatalytic and radiosensitization therapies aimed at eradicating primary and metastatic tumors. Through comprehensive experimental data, theoretical calculations, and in situ FTIR experiments, the authors elucidated the catalytic mechanism of the unique Mn-organic ligands in producing ROS. They also demonstrated that MnBTC-Ru could enhance the sensitivity to X-ray irradiation, increase DNA damage in cancer cells, and, when combined with anti-PD-1 therapy, promote robust systemic antitumor responses while preventing tumor recurrence and metastasis. The study was comprehensive, its concept is new, and its figures were well prepared. Nevertheless, the authors may take into account the following aspects in revising the paper.”

Response to the general comment:

We sincerely appreciate your constructive feedback on our manuscript. Your comments and suggestions have been invaluable in enhancing the overall quality of our work. In response, we have conducted a comprehensive revision of the manuscript based on your queries and concerns. We believe that these significant revisions have enhanced the clarity of the mechanistic aspects and the quality of this paper. We hope you will agree with this revision, and we thank you once again for your considerable efforts.

(1) *About the simulation of the natural Mn-POD active center in MnBTC-Ru, what advantages did the addition of Ru offer to radiosensitization therapy? How is Ru in comparison with Os or Ir?*

Response to comment:

Thanks for your important and helpful comments on improving the quality of our manuscript. Regarding the advantages of Ru addition in radiosensitization therapy: firstly, as a member of the iron (Fe) group elements, Ru exhibits high biocompatibility comparable to that of Fe, rendering it suitable for application in radiosensitization and other various biomedical applications. Secondly, the relatively high atomic number of Ru ($Z = 44$) facilitates increased X-ray absorption and the production of secondary electrons *via* the photoelectric effect and Compton scattering, thereby directly inducing DNA ionization in tumor cells. Moreover, in comparison to Fe, Ru possesses a greater number of *d*-electrons, an ample quantity of unoccupied orbitals, and superior redox stability. These electronic properties facilitate bond formation in biocatalytic processes, enable effective coordination of reaction intermediates, and positively modulate the adsorption and dissociation kinetics involved in biocatalysis, thereby enhancing O_2 generation from H_2O_2 substrates and amplifying radiation-induced ROS production. We have elaborated on the role of Ru more clearly in the revised manuscript, as also outlined below:

Page 3 in the revised manuscript: “Recently, our research team reported that ruthenium (Ru), a relatively high atomic number metal belonging to the iron group, possesses good biocompatibility and unique biocatalytic properties—specifically, more *d* electrons, sufficient unoccupied orbitals, and superior redox stability—that enable efficient O_2 generation through rapid electron transfer with H_2O_2 as the substrate.”

Page 10 in the revised manuscript: “We propose that the biocatalytic MnBTC-Ru complex exhibits exceptional dual functionality in generating both O_2 and ROS; meanwhile, it can produce ROS in significant quantities under X-ray radiation, thereby enhancing radiosensitization, reducing radiation resistance, and improving its overall antitumor efficacy compared to the control biocatalyst (C-Ru) (Fig. 3a).”

Regarding the comparison of Ru with Os and Ir, the selection of the optimal precious metal for our experiments required a comprehensive evaluation of several critical factors, including radiosensitization efficacy, biocompatibility, enzyme-mimetic activities (encompassing the overall generation of ROS and O_2), and alignment with the specific therapeutic objectives of our study. Based on a thorough assessment of these criteria, Ru was chosen for its superior and well-balanced

performance across all relevant parameters. A detailed discussion of this selection will be presented from the following two perspectives.

1) Regarding biosafety, we conducted a comprehensive evaluation of multiple factors to achieve an optimal balance in element selection. Although elements with higher atomic numbers, such as Os and Ir, theoretically exhibit greater X-ray absorption, their relatively high biological toxicity limits their clinical applicability. Our preliminary experiments demonstrated that Ru offers significantly better biocompatibility than Os and Ir. While Os and Ir showed slightly superior radiosensitization effects compared to Ru, considerations of biological safety led us to select Ru as the preferred element.

2) The enzyme-mimetic activity of Ru is notably well-balanced. In selecting radiotherapy sensitizers, it is important not only to consider the radiosensitization efficacy—which generally improves with increasing atomic number—but also to assess the balance of their enzyme-mimetic activities. Ru offers distinct advantages in this regard, as its enzyme-mimetic activity maintains an optimal balance between ROS generation and O₂ production, thereby facilitating both tumor cell killing and immune cell activation. However, our preliminary experiments indicate that Ir preferentially promotes O₂ production, while Os enhances ROS production; this imbalance renders them less suitable to meet the demands of our therapeutic system.

We fully acknowledge the reviewers' viewpoint that Os and Ir remain promising candidates as radiotherapy sensitizers, warranting further investigation. In future studies, we plan to focus on optimizing both the design and performance balance of Os and Ir. We sincerely appreciate these valuable suggestions, which provide important guidance for advancing our research.

(2) It would be beneficial to include further DLS testing to assess the dispersibility of MnBTC-Ru, as this will help determine any potential adverse effects in vivo.

Response to comment:

We appreciate your thorough review and valuable recommendations regarding the dispersion assessment of MnBTC-Ru. We fully recognize the importance of dynamic light scattering (DLS) analysis in evaluating material dispersion and its implications for potential *in vivo* safety.

As you noted, DLS provides critical insights into material dispersion. Effective dispersion minimizes nonspecific nanoparticle aggregation in biological systems, thereby reducing the risk of adverse effects. In response to your suggestion, we conducted a systematic DLS analysis of MnBTC-Ru under the following conditions: phosphate-buffered saline (PBS), 25 °C, with three replicates. The results indicate an average hydrodynamic diameter of 578 nm, closely matching the inherent particle size of MnBTC-Ru, confirming its favorable dispersion characteristics (Supplementary Fig. 4). Additionally, DLS tests in mouse serum showed a particle size of 525 nm, further indicating that MnBTC-Ru maintains dispersion without significant aggregation in a simulated physiological environment.

We sincerely appreciate your valuable input in enhancing the quality of our manuscript. The corresponding details have been incorporated into the revised manuscript and revised supplementary information, as also outlined below:

Page 5 in the revised manuscript: “Dynamic light scattering analysis indicates that MnBTC-Ru maintains good dispersion in both phosphate-buffered saline (PBS) and serum culture medium, with average hydrodynamic diameters of 578 nm and 525 nm, respectively (Supplementary Fig. 4). These values closely correspond to the inherent particle size of MnBTC-Ru, demonstrating favorable dispersibility under simulated physiological conditions and supporting its low potential for *in vivo* aggregation and related adverse effects.”

Supplementary Fig. 4. Dynamic light scattering (DLS) analysis of MnBTC-Ru conducted in phosphate-buffered saline (PBS) and serum-containing culture medium ($n = 3$ independent experiments, data are presented as mean \pm SD). Source data are provided as a Source Data file.

(3) While *in situ* infrared spectroscopy is a valuable characterization method, validating these findings against existing literature would enhance the analysis, some references should be added to validate the peaks.

Response to comment:

Thank you for your constructive feedback and valuable suggestions. We recognize that robust literature support is vital for substantiating the attribution of infrared peaks. In response to your remarks, we have incorporated additional references that provide a comprehensive context for the intermediates indicated by the *in-situ* infrared spectra. The revised manuscript now includes the earlier relevant literature, which we believe enhances the clarity and depth of our findings.

Page 11 in the revised manuscript: “*In-situ* FTIR spectroscopy has also been employed to examine the structural changes of the ROS intermediates during the peroxidase-like process, indicating the formation of $\bullet\text{O}_2^-$ and $\ast\text{OOH}$ intermediates (Fig. 3k, l)⁵⁰⁻⁵².”

Reference:

- 50 Yue, J.-Y. et al. Thiophene-Containing Covalent Organic Frameworks for Overall Photocatalytic H₂O₂ Synthesis in Water and Seawater. *Angew. Chem. Int. Ed.* 62, e202309624 (2023).
- 51 Wang, X. et al. Ambient Preparation of Benzoxazine-based Phenolic Resins Enables Long-term Sustainable Photosynthesis of Hydrogen Peroxide. *Angew. Chem. Int. Ed.* 62, e202302829 (2023).
- 52 Liu, R. et al. Linkage-engineered donor-acceptor covalent organic frameworks for optimal photosynthesis of hydrogen peroxide from water and air. *Nat. Catal.* 7, 195-206 (2024).

(4) The authors have made progress in verifying the mechanism of $\ast\text{OOH}$ interaction between different material centers; however, further clarification on why $\ast\text{OOH}$ serves as a key intermediate in forming $\bullet\text{O}_2^-$ and $^1\text{O}_2$ would be helpful.

Response to comment:

Thank you for your valuable comments and suggestions. Your recommendation to clarify why $\ast\text{OOH}$ acts as a key intermediate in the generation of $\bullet\text{O}_2^-$ and $^1\text{O}_2$ provides important guidance for refining the reaction mechanism. Below, we systematically address this issue by integrating our experimental data with relevant literature evidence.

1) Regarding *OOH as a key intermediate in the formation of •O₂⁻: the dynamic equilibrium between *OOH adsorbed on the catalyst surface and protons (H⁺) in solution during catalysis constitutes the central mechanism for the generation of •O₂⁻, represented by the equilibrium reaction: OOH ↔ •O₂⁻ + H⁺ (Fig. R1). This equilibrium process has been extensively validated across electrocatalytic, photocatalytic, and enzymatic catalysis systems (*PNAS* **2024**, 33, e2407012121; *Nano Res.* **2018**, 11, 4955-4984; *Adv. Mater.* **2022**, 34, 2108646).

2) Regarding *OOH as a key intermediate in the formation of ¹O₂: the generation of ¹O₂ depends on the further transformation of *OOH. According to the Russell mechanism, two adsorbed superoxide radicals (*OOH or •O₂⁻) can generate ¹O₂ through a recombination reaction (Fig. R2). We have cited the recent authoritative literature on the application of the Russell mechanism in nanocatalyst systems (*J. Am. Chem. Soc.* **2023**, 145, 8965-8978; *Angew. Chem. Int. Ed.* **2019**, 58, 9846-9850), in order to enhance the universality of the theoretical basis. Furthermore, the generation of *OOH can be observed by *in-situ* infrared spectroscopy (Fig. 3k, l). Our theoretical calculations further indicate that the electron interaction between Ru and *OOH in MnBTC-Ru is relatively weak (Fig. 3m, n), and the •O₂⁻ and ¹O₂ released after dissociation can be directly detected by electron paramagnetic resonance (EPR) (Fig. 3i, j). Therefore, by monitoring the generation and dissociation behavior of *OOH, the release capacity of •O₂⁻ and ¹O₂ of the MnBTC-Ru can be indirectly evaluated.

Thank you once again for your insightful feedback. Your thorough review has greatly improved the logical coherence and scientific rigor of our study. The corresponding details have been incorporated into the revised manuscript, as also outlined below:

Page 11 in the revised manuscript: “Considering that, during the catalytic process, the dynamic equilibrium between *OOH adsorbed on the material surface and protons (H⁺) in solution constitutes the central mechanism to the generation of •O₂⁻, and that the formation of ¹O₂ similarly relies on the transformation of *OOH *via* the Russell mechanism⁵³⁻⁵⁷, we, therefore, regard *OOH as an ideal and essential model for a deeper investigation into the intrinsic relationship between the structure of the Ru cluster coordinated with Mn-organic ligands and its catalytic properties through DFT calculations.”

Reference:

- 53 Li, L. *et al.* Modulating Electron Transfer in Vanadium-Based Artificial Enzymes for Enhanced ROS-Catalysis and Disinfection. *Adv. Mater.* **34**, 2108646 (2022).

- 54 Wang, C. *et al.* Specific Generation of Singlet Oxygen through the Russell Mechanism in Hypoxic Tumors and GSH Depletion by Cu-TCPP Nanosheets for Cancer Therapy. *Angew. Chem. Int. Ed.* **58**, 9846-9850 (2019).
- 55 Li, Y., Zhang, D., Wang, P., Qu, J. & Zhan, S. Superoxide radicals mediated by high-spin Fe catalysis for organic wastewater treatment. *PNAS* **121**, e2407012121 (2024).
- 56 Liu, Y. *et al.* Multi-enzyme Co-expressed Dual-Atom Nanozymes Induce Cascade Immunogenic Ferroptosis via Activating Interferon- γ and Targeting Arachidonic Acid Metabolism. *J. Am. Chem. Soc.* **145**, 8965-8978 (2023).
- 57 Ferreira, C. A., Ni, D., Rosenkrans, Z. T. & Cai, W. Scavenging of reactive oxygen and nitrogen species with nanomaterials. *Nano Res.* **11**, 4955-4984 (2018).

***OOH serves as a key intermediate in the formation of $\bullet\text{O}_2^-$**

Fig. R1 Literature on $\bullet\text{OOH}$ being a key intermediate for the formation of $\bullet\text{O}_2^-$.

***OOH serves as a key intermediate in the formation of $^1\text{O}_2$**

J. Am. Chem. Soc. **2023**, 145, 8965-8978

During the treatment, FeCo/Fe-Co DAzyme/PL can convert over-expressed H_2O_2 in tumor tissue to O_2 , and O_2 could further combine with the free AA released from PLA2-catalyzed phospholipids to generate AA-O-OH under LOX catalysis. Finally, AA-O-OH can generate a large amount of singlet oxygen ($^1\text{O}_2$) through the Russell mechanism. Mean-

Angew. Chem. Int. Ed. **2019**, 58, 9846-9850

syntheses will significantly inhibit their clinical translation. In essence, $^1\text{O}_2$ can be generated from reactions of biological hydroperoxides in the presence of a trace amount of metal ion or enzyme according to the Russell mechanism.^[5] More importantly, most biological hydroperoxides can be produced through the peroxidation of reactive oxygen species (ROS).^[5]

Fig. R2 Literature on *OOH being a key intermediate for the formation of $^1\text{O}_2$.

Fig. 3 **k** *In-situ* FTIR spectrum and **l** the corresponding contour plot of MnBTC-Ru for the peroxidase-like process.

Fig. 3 **m** Differential charge density analysis of Ru centers (yellow and cyan represent charge accumulation and depletion, respectively, with a cutoff value of $0.005 \text{ e} \cdot \text{Bohr}^{-3}$ for the density-difference isosurface). Ru, yellow; Mn, blue; C, khaki; H, white; O, purple. **n** Calculated Bader charge of MnBTC-Ru and C-Ru with the adsorption of a *OOH intermediate.

Fig. 3 EPR spectra for recording the **i** $\bullet\text{O}_2^-$ signal and the **j** $^1\text{O}_2$ signal.

(5) Two important issues regarding the biological mechanisms of this system are missing. First, more in-depth analysis is needed on the role of this system in activating antitumor immunity, especially given the gene sequencing data; second, how is the selectivity of this system when acting on tumor cells against normal cells?

Response to comment:

We sincerely appreciate your constructive feedback and valuable suggestions. This study demonstrates that MnBTC-Ru-mediated activation of systemic antitumor immune responses represents the pivotal mechanism underlying its efficacy as a highly potent radiosensitizer for achieving systemic tumor eradication. *In vitro* experiments demonstrate that the RT+MnBTC-Ru significantly induces immunogenic cell death (ICD) in tumor cells, characterized by the robust release

of damage-associated molecular patterns (DAMPs), including ATP and calreticulin. Functioning as critical danger signals, these DAMPs not only potently activate multiple immune cell populations but also substantially reverse the immunosuppressive status of the tumor microenvironment (*Immunity*, **2024**, 57, 752-771; *Annu. Rev. Pathol.* **2020**, 15, 493-518.). *In vivo*, we systematically evaluated the radiosensitizing effects and immunomodulatory mechanisms of MnBTC-Ru in bilateral subcutaneous tumor models and immune memory animal models. Key experimental findings include: 1) Flow cytometry analysis reveals a significant increase in the proportion of CD8⁺CD69⁺ T cells in tumors treated with RT+MnBTC-Ru. As an activated CD8⁺ T cell subset, CD8⁺CD69⁺ T cells suggest that RT+MnBTC-Ru may enhance tumor clearance by potentiating CD8⁺ T cell cytotoxicity (*Int. Immunol.* **2022**, 34, 555-561); 2) Transcriptome sequencing demonstrates significant activation of immune-related pathways, including immune response and immune system process, further confirming the pivotal role of RT+MnBTC-Ru in activating antitumor immunity; 3) Immunofluorescence (IF) analysis of pathological sections from distant tumor-draining lymph nodes showed that RT+MnBTC-Ru combined with anti-PD-1 treatment markedly promoted CD8⁺ T cell proliferation within lymph nodes. These results demonstrate that MnBTC-Ru not only enhances local radiotherapy efficacy but also induces systemic antitumor immune responses, thus leading to systemic tumor regression.

Building upon your insightful recommendations, we performed an in-depth analysis of the transcriptomic sequencing data. First, we conducted protein-protein interaction (PPI) network analysis on immune-related differentially expressed genes (Supplementary Fig. 22). A PPI network was constructed with IFNG, TNF, IL6, and PTPRC as central hubs, highlighting their pivotal roles in immunoregulation. IFNG encodes interferon-gamma (IFN- γ), a master regulator of immune responses that orchestrates host defense by activating diverse immune cell populations, enhancing antimicrobial capacity, and modulating immune signaling pathways (*Sci. Immunol.* **2024**, 9, 4893; *Nat. Genet.* **2007**, 39, 1453–1460). TNF, which encodes tumor necrosis factor-alpha (TNF- α), contributes to the formation of a pro-inflammatory tumor microenvironment by activating immune cells and plays a central role in antitumor immunity by inducing apoptosis in tumor cells through synergistic immune activation (*Nat. Rev. Immunol.* **2023**, 23, 289–303; *Immunity* **2025**, 58, 585–600). IL6 encodes interleukin-6 (IL-6), a pleiotropic cytokine that promotes immune surveillance and suppresses tumor growth by enhancing the activity of cytotoxic T lymphocytes and natural killer cells. PTPRC, which

encodes the transmembrane protein tyrosine phosphatase CD45, is essential for T cell differentiation and functions as a key regulator of T cell maturation and immune homeostasis (*Nat. Genet.* **2024**, 56, 2739–2752; *Annu. Rev. Immunol.* **1992**, 10, 645–674). These findings provide critical mechanistic insights into how MnBTC-Ru enhances intratumoral T cell infiltration and underscore the central role of these immunoregulatory genes in orchestrating antitumor immune responses.

To evaluate the effects of MnBTC-Ru on the tumor immune microenvironment, we performed a quantitative analysis of immune cell subtypes using the ImmuCellAI algorithm, based on RNA-seq transcriptomic data. Compared to the control group (median score: 1.663) and the RT group (median score: 2.680), the combination treatment with RT+MnBTC-Ru results in a notably higher overall immune infiltration score (median score: 3.492; Supplementary Fig. 23). The increase in immune infiltration implies that MnBTC-Ru may facilitate immune cell recruitment, potentially by improving antigen presentation or alleviating immunosuppressive conditions in the tumor microenvironment. Supporting this, further analysis (Supplementary Fig. 24) shows a greater presence of immune cells with antitumor activity in the RT+MnBTC-Ru group, along with a decline in suppressive cells such as M2 macrophages. This shift toward a more active immune profile may help explain the improved therapeutic response observed with the RT+MnBTC-Ru treatment.

Pathway enrichment analysis based on the Kyoto Encyclopedia of Genes and Genomes (KEGG) database (Supplementary Fig. 25) reveals prominent activation of signaling pathways in the RT+MnBTC combination treatment group, primarily involving redox regulation, immune modulation, and inflammation–apoptosis signaling. Specifically, pathways related to oxidative phosphorylation and the HIF-1 signaling pathway were enriched, suggesting disrupted redox homeostasis and hypoxia adaptation (*Nat. Rev. Cancer* **2003**, 3, 721–732; *Cell Metab.* **2021**, 33, 2398–2414). Meanwhile, the enrichment of cytokine–cytokine receptor interaction and T cell receptor signaling pathways indicates robust immune activation. Furthermore, activation of the NF-kappa B and TNF signaling pathways suggests engagement of inflammatory and pro-apoptotic programs (*Nat. Rev. Immunol.* **2023**, 23, 289–303). These results suggest that MnBTC-Ru enhances radiosensitivity through coordinated modulation of oxidative stress, immune response, and cell death pathways. It disrupts ROS homeostasis in tumor cells, leading to irreversible oxidative damage; promotes antitumor immunity via activation of cytokine–T cell signaling cascades; and reprograms the NF-κB/TNF signaling axis to drive tumor cells

toward a death-sensitive state. This multi-target synergy ultimately induces programmed cell death.

In response to the second point raised in the review, we believe that the favorable cellular selectivity of MnBTC-Ru as a radiosensitizer is primarily attributed to two factors: the differential sensitivity of normal versus tumor cells to reactive oxygen species (ROS) and the distinct roles of oxygen in normal tissues versus the tumor microenvironment. To counteract ROS-induced damage, normal cells have evolved intricate antioxidant defense systems that maintain redox homeostasis. Various enzymatic and non-enzymatic antioxidants, such as superoxide dismutase (SOD), are compartmentalized within different cellular organelles to mitigate oxidative stress (*Med. Res. Rev.* **2024**, 44, 1566–1595). In contrast, tumor cells typically exhibit elevated basal ROS levels, structural instability, and compromised antioxidant systems, rendering them highly susceptible to oxidative damage. Acute exposure to high ROS concentrations can trigger oxidative stress that irreversibly damages proteins, lipids, and DNA, ultimately leading to tumor cell apoptosis or necrosis. Normal cells, however, are generally capable of withstanding comparable ROS levels without undergoing cell death (*Nat. Rev. Drug Discov.* **2009**, 8, 579–591; *Nat. Rev. Drug Discov.* **2013**, 12, 931–947). Leveraging this differential sensitivity, MnBTC-Ru selectively eradicates tumor cells by rapidly generating cytotoxic ROS while sparing normal tissues. Moreover, the hypoxic nature of solid tumors further enhances the tumor selectivity of MnBTC-Ru. Hypoxia is a hallmark of tumor heterogeneity and a key driver of therapeutic resistance, particularly in RT, where oxygen is essential for stabilizing radiation-induced DNA damage (*Signal Transduct Target Ther.* **2023**, 8, 70). Ionizing radiation generates free radicals that induce DNA lesions, and the presence of molecular oxygen is required to "fix" these radicals into irreversible adducts—a process known as the oxygen fixation effect. In hypoxic conditions, this process is impaired, allowing tumor cells to repair DNA damage and develop radioresistance (*Nat. Rev. Cancer* **2025**, 25, 167–188). Owing to its catalytic properties, MnBTC-Ru can continuously generate molecular oxygen in situ within the tumor microenvironment. These oxygen molecules diffuse into hypoxic tumor regions, bind to radiation-induced DNA radicals, and prevent their repair, thereby restoring radiosensitivity and enhancing the specificity of tumor cell killing.

The corresponding new results have been included in the revised manuscript and supporting information, which are also shown below:

Page 20 in the revised manuscript: “Protein–protein interaction (PPI) network analysis of immune-related differentially expressed genes (Supplementary Fig. 22) shows that the RT+MnBTC-Ru combination treatment promotes the formation of a tightly connected network centered on key immune regulators, including IFNG (encoding interferon- γ), TNF (encoding tumor necrosis factor), IL6 (encoding interleukin-6), and PTPRC (encoding CD45 antigen). The strong interactions among these core molecules suggest that the combination therapy reshapes the tumor immune microenvironment by coordinating multiple immune pathways at the molecular level. To further investigate the immunomodulatory effects of RT+MnBTC-Ru, we estimate the composition of immune cell populations within the TME using RNA-seq transcriptomic data analyzed with the ImmuCellAI algorithm. The RT+MnBTC-Ru group shows a significantly higher immune infiltration score (3.492) compared to both the control (1.663) and RT (2.680) groups (Supplementary Fig. 23). This increase was primarily driven by a rise in antitumor immune effector cells accompanied by a reduction in immunosuppressive populations, such as M2 macrophages (Supplementary Fig. 24). Such a favorable shift in the immune cell landscape may help explain the improved therapeutic outcomes observed with the RT+MnBTC-Ru treatment. Consistent with these findings, Gene Ontology (GO) functional enrichment analysis reveals that these differentially expressed genes were significantly enriched in immune–related biological processes (Fig. 6f). In addition, pathway enrichment analysis based on the Kyoto Encyclopedia of Genes and Genomes (KEGG) database (Supplementary Fig. 25) revealed pronounced activation of signaling pathways in the RT+MnBTC combination treatment group, predominantly associated with redox regulation, immune modulation, and inflammation-apoptosis signaling.”

Supplementary Fig. 22. Protein-protein interaction (PPI) network of immune-related differentially expressed genes. Source data are provided as a Source Data file.

Supplementary Fig. 23. The immune infiltration score estimated based on gene expression profiles using ImmuCellAI, reflects the overall level of immune activity, with higher scores indicating stronger immune cell infiltration (n = 3 independent experiments, data are presented as mean \pm SD). Source data are provided as a Source Data file.

Supplementary Fig. 24. Heatmap showing the relative abundance of immune cell types deconvoluted from RNA-seq data. Source data are provided as a Source Data file.

Supplementary Fig. 25. KEGG enrichment analysis of upregulated genes in the RT+MnBTC-Ru group. Source data are provided as a Source Data file.

(6) More discussion from a clinical translation perspective is preferred on the current limitations of tumor radiotherapy and how this system could potentially tackle these issues and accelerate clinical application.

Response to comment:

We sincerely appreciate your insightful comments and constructive suggestions, which have significantly enhanced our discussion regarding the clinical translation potential of this study. In response to your valuable input, we present our response below, accompanied by the corresponding revisions made to the manuscript.

Radiotherapy, which uses high-energy X-rays to induce localized tumor destruction through double-strand DNA breaks and the generation of reactive oxygen species (ROS), remains a cornerstone

in the treatment of solid tumors, often in combination with chemotherapy or immunotherapy (*Lancet Oncol.* **2019**, 20, e434). However, both intrinsic and acquired resistance to radiation significantly limit its clinical efficacy (*Mol. Cancer* **2023**, 22, 96). Tumor radioresistance arises from multiple factors, including enhanced DNA repair mechanisms in cancer cells (*Nat. Genet.* **2001**, 27, 247; *Mol. Cell* **2017**, 66, 801), persistent hypoxia in the tumor microenvironment (*Semin. Cancer Biol.* **2023**, 96, 5), and the establishment of an immunosuppressive milieu (*Cancer Cell* **2021**, 39, 725). Together, these factors reduce the responsiveness of tumors to conventional RT, underscoring the need for more effective radiosensitization strategies. Despite extensive efforts, traditional radiosensitizers such as metronidazole, misonidazole, and etanidazole have shown limited clinical success when combined with RT (*Oncol. Res.* **1994**, 6, 509; *Int. J. Radiat. Oncol. Biol. Phys.* **1984**, 10, 425). These agents lack tumor selectivity and are associated with systemic toxicity. Moreover, the currently developed radiosensitizing materials (e.g., lanthanides, hafnium, and gold) are restricted in future clinical translation due to their insufficient tumoricidal effects and low biodegradability-induced chronic toxicities (*Mol. Cancer* **2023**, 22, 94; *J. Nanobiotechnol.* **2020**, 18, 122).

To address these challenges, we have designed a novel and bioinspired radiosensitizer, MnBTC-Ru, which integrates catalytic ruthenium clusters within Mn-organic ligands. This structure enables a synergistic therapeutic approach that combines biocatalytic ROS generation with radiotherapy. MnBTC-Ru effectively induces immunogenic cell death, promoting the release of damage-associated molecular patterns and facilitating immune cell recruitment to the tumor site following radiotherapy. Importantly, when combined with anti-PD-1 therapy, MnBTC-Ru elicits a robust systemic antitumor immune response, suppressing both primary and metastatic tumors and reducing recurrence. By transforming RT into an immunostimulatory “adjuvant” through coordinated ROS production and immune activation, MnBTC-Ru expands the therapeutic window of RT and enhances the efficacy of immunotherapy. In addition to enhancing radiosensitivity and remodeling the immunosuppressive tumor microenvironment, MnBTC-Ru significantly improves tumor antigenicity. Its high biocompatibility and biodegradability further support its potential for clinical translation, offering a promising strategy for safer and more effective tumor radioimmunotherapy.

Page 25 in the revised manuscript: “Radiotherapy remains a cornerstone in the treatment of solid tumors, often in combination with chemotherapy or immunotherapy⁷⁸. However, both intrinsic and

acquired resistance to radiation significantly limits its clinical efficacy⁷⁹. Tumor radioresistance arises from multiple factors, including enhanced DNA repair mechanisms in cancer cells^{80,81}, persistent hypoxia in the tumor microenvironment⁸², and the establishment of an immunosuppressive milieu⁸³. Together, these factors reduce the responsiveness of tumors to conventional RT, underscoring the need for more effective radiosensitization strategies. Despite extensive efforts, traditional radiosensitizers such as metronidazole, misonidazole, and etanidazole have shown limited clinical success when combined with RT^{84,85}. These agents lack tumor selectivity and are associated with systemic toxicity. Moreover, the currently developed radiosensitizing materials (e.g., lanthanides, hafnium, and gold) are restricted in future clinical translation due to their insufficient tumoricidal effects and low biodegradability-induced chronic toxicities^{15,17}.

Our proof-of-concept design paves the way for the development of radiosensitizers that can synergize biocatalytic ROS production with defensive system activation, transforming RT into an effective "adjuvant" that enhances antitumor responses, thereby expanding the scope of patients who can benefit from both RT and checkpoint inhibitor therapy. The TME is crucial in systemic antitumor responses, with various cells playing key roles in this complex ecosystem⁸⁶. Combining MnBTC-Ru with RT enhances the recruitment of dendritic cells by providing more DAMPs through ICD, improving the therapeutic outcomes of RT⁸⁷. Here, we have developed MnBTC-Ru as a TME-adaptive and biocatalytic radiosensitizer for enhancing radiosensitivity, modulating the TME, and enhancing tumor antigenicity.

The innovative integration of ROS biocatalysis, oxygen sensitization, and bioinspired metal-organic coordination-based radiosensitizers presents a promising avenue for advancing biocatalytic and radiotherapeutic strategies in malignant tumors. This approach holds significant potential, particularly in preventing tumor metastasis and recurrence. To realize its full clinical potential, future research should prioritize systematically evaluating the long-term biocompatibility and biodegradability of these biocatalytic and radioactive materials. Additionally, a deeper understanding of their mechanisms of action within complex biological environments is essential to optimize their therapeutic application. These efforts will not only provide a robust scientific basis for uncovering the full scope of their therapeutic capabilities but also accelerate their effective translation into clinical use. By bridging basic research and clinical implementation, this direction has the potential to revolutionize

cancer treatment, offering patients more precise and effective therapeutic solutions.”

Reference:

- 78 Pitroda, S. P., Chmura, S. J. & Weichselbaum, R. R. Integration of radiotherapy and immunotherapy for treatment of oligometastases. *Lancet Oncol.* **20**, e434-e442 (2019).
- 79 Wu, Y., Song, Y., Wang, R. & Wang, T. Molecular mechanisms of tumor resistance to radiotherapy. *Mol. Cancer* **22**, 96 (2023).
- 80 Khanna, K. K. & Jackson, S. P. DNA double-strand breaks: signaling, repair and the cancer connection. *Nat. Genet.* **27**, 247-254 (2001).
- 81 Blackford, A. N. & Jackson, S. P. ATM, ATR, and DNA-PK: The Trinity at the Heart of the DNA Damage Response. *Mol. Cell* **66**, 801-817 (2017).
- 82 Semenza, G. L. Targeting intratumoral hypoxia to enhance antitumor immunity. *Semin. Cancer Biol.* **96**, 5-10 (2023).
- 83 Li, M. O. *et al.* Innate immune cells in the tumor microenvironment. *Cancer Cell* **39**, 725-729 (2021).
- 84 Overgaard, J. Clinical evaluation of nitroimidazoles as modifiers of hypoxia in solid tumors. *Oncol. Res.* **6**, 509-518 (1994).
- 85 Brown, J. M. Clinical trials of radiosensitizers: what should we expect? *Int. J. Radiat. Oncol. Biol. Phys.* **10**, 425-429 (1984).
- 86 Binnewies, M. *et al.* Understanding the tumor immune microenvironment (TIME) for effective therapy. *Nat. Med.* **24**, 541-550 (2018).
- 87 Baumann, M., Krause, M. & Hill, R. Exploring the role of cancer stem cells in radioresistance. *Nat. Rev. Cancer* **8**, 545-554 (2008).

Reviewer #2 (Remarks to the Author):

“The paper «Bioinspired Ruthenium-Manganese-Oxygen Complex for Biocatalytic and Radiosensitization Therapies to Eradicate Primary and Metastatic Tumors» has developed a novel MOF antitumor material, which has certain practical significance for tumor treatment. However, there are still some problems, mainly as follows:”

Response to the general comment:

We sincerely appreciate your acknowledgment of our research efforts. Based on your comments and the suggestions from other reviewers, we have conducted more systematic experiments and refined the content throughout the manuscript. All necessary data have been added to support our claims, and we have thoroughly addressed all questions and concerns in the revised manuscript and supporting information. Therefore, we believe that the quality of this paper has been significantly enhanced. We hope you will agree with this revision, and we thank you once again for your considerable efforts.

(1) *The characterization results of MnBTC are inconsistent with those reported in this literature, especially XRD. Please explain the reasons. Chen, Y., Tian, Q., Wang, H. Y., Ma, R. N., Han, R. T., Wang, Y., Ge, H. B., Ren, Y. J., Yang, R., Yang, H. M., Chen, Y. J., Duan, X. Z., Zhang, L. B., Gao, J., Gao, L. Z., Yan, X. Y., & Qin, Y. (2022). A Manganese-Based Metal-Organic Framework as a Cold-Adapted Nanozyme. *Advanced Materials*.*

Response to comment:

We sincerely appreciate your thorough and detailed review of the material characterization presented in our manuscript. In response to your comments concerning the XRD analysis of the MnBTC material, we have re-conducted comprehensive experimental verification and data analysis.

Upon review, we observed that the synthetic conditions employed in our study differ from those reported in the literature, which results in variation in crystallinity compared to the literature. We conducted a detailed analysis of the initial XRD pattern and identified minor crystalline peaks at 10.4° and 20.9°, consistent with previously reported literature. However, the equipment used in the initial analysis (DX-2700BH, HaoYuan Instrument, China) exhibited limited sensitivity to diffraction signals from low-crystallinity materials. Additionally, the use of glass slides for powder sample mounting generated a pronounced amorphous scattering peak near 25°, which likely suppressed adjacent crystalline diffraction signals, particularly the peak at 20.9°. To further validate the material's structure, we have reanalyzed the same samples using a new instrument (Ultima IV, Rigaku, Japan) with optimized test parameters, including a scanning speed of 3°/min and a quartz sample stage. This improved setup revealed distinct MOF crystallization peaks at 10.4°, 20.9°, and 29.1°, in agreement with the mentioned literature (Supplementary Fig. 1 and Fig. R3).

These new results confirm the successful synthesis of the MnBTC materials. The initial discrepancies in characterization were primarily attributable to differences in testing methodologies and XRD instruments rather than the materials themselves. We have updated all relevant data and provided detailed testing conditions in the revised supplementary information. We sincerely appreciate your constructive suggestions, which have significantly contributed to the enhancement of our material characterization efforts.

Page 5 in the revised manuscript: “X-ray diffraction analysis confirms the successful synthesis of crystalline MnBTC support (Supplementary Fig. 1).”

Page 20 in the revised supporting information: “X-ray diffraction (Ultima IV, Rigaku, Japan) was used to analyze the crystal structures of the catalysts, employing Cu K α radiation over a 2θ range of 5-30°, with a scanning speed of 3°/min and a quartz sample stage.”

Current XRD pattern: detected using the
Ultima IV, Rigaku, Japan

Supplementary Fig. 1. X-ray diffraction patterns of MnBTC-Ru and MnBTC. Source data are provided as a Source Data file.

XRD pattern reported in the literature

Adv. Mater. **2024**, 36, 2206421

Fig. R3 XRD pattern of MnBTC reported in the mentioned literature.

(2) *What is the basis for drawing the structure of Figure 1b? There should be Mn³⁺ (referred to as Mn⁴⁺ in the text) here, otherwise how is ROS generated? This structural diagram should not reflect the oxidation effect of Mn ions.*

Response to comment:

Thank you for your thorough review and insightful suggestions concerning the valence state of Mn in our manuscript. The inquiry you posed regarding the structural foundation depicted in Fig. 1b is particularly important. In response, we have refined the structural illustration to provide a clearer explanation of the design rationale and the key structural features of the material system.

Regarding the structural design presented in Fig. 1b, we indeed need to clarify its scientific basis more clearly. The natural Mn-peroxidase enzyme operates as a dynamic coordination system in which variations in the distances between coordinating amino acids, together with the synergistic influence of the Fe auxiliary center, collaboratively modulate the redox properties of the active site to achieve efficient catalytic activity. In the MnBTC-Ru material system, we have established a multi-level cooperative regulation system to control the valence state of Mn ions. Firstly, the Mn ion establishes a stable coordination environment by binding to the oxygen atom of the BTC ligands, which in turn determines its fundamental oxidation state of Mn⁺³. Simultaneously, the incorporated Ru nanoclusters

perform several critical functions within the system: 1) acting as efficient X-ray absorbers that enhance absorption and generate secondary electrons; 2) their metallic nature facilitates electron transfer through Ru-Mn bonding, thereby dynamically modulating the valence state of Mn. This synergistic interaction is similar to the regulatory mechanisms observed in natural enzyme systems, where the active site is collaboratively controlled by ligands and associated cofactors. Furthermore, the active center of RuMn-O-C in MnBTC-Ru closely resembles that of the Mn-O-C center in natural Mn-peroxidase. Notably, our system also exhibits an external field response, allowing the overall valence state to be tuned by adjusting the X-ray irradiation conditions. This multi-scale, multi-mechanism cooperative regulation underpins the superior redox activity demonstrated by the MnBTC-Ru system.

As the reviewer has correctly noted, Mn³⁺ is a key component of our enzyme-mimicking system, and the presence of Mn⁴⁺ can be definitively excluded. In response to your suggestions, we have revised Fig. 1b and supplemented the Supporting Information with additional diagrams to more clearly illustrate the structural features of the MnBTC-Ru (Fig. 1b and Supplementary Fig. 7). It is important to emphasize that Fig. 1b primarily aims to depict the synergistic mechanism of the active centers, rather than presenting the comprehensive structural details of the material. **Within the MnBTC-Ru system, Mn exists in multiple valence states: 1)** In a typical MnO₆ coordination environment, where Mn is not bonded to Ru and is coordinated exclusively to the oxygen atoms of the BTC ligands, the Mn valence state remains stably at +3. **2)** In cases of coordination unsaturation—where Mn is similarly not bonded to Ru—coordination defects result in a reduction of the Mn valence state to +2. **3)** The electron transfer between the RuMn active centers—formed by the addition of Ru nanoclusters—and the influence of external X-ray fields induce Mn to exhibit valence states ranging from +2 to +3, where Mn²⁺ corresponds to electron gain and Mn³⁺ to the absence thereof; this core mechanism is illustrated in Fig. 1b. This multi-tiered valence state regulation mechanism, encompassing both the intrinsic effects of the coordination environment and the modulation induced by the additional incorporation of Ru, collectively establishes the distinctive coexistence of Mn²⁺ and Mn³⁺ valence states within the MnBTC-Ru.

We sincerely appreciate your valuable input, which has greatly enhanced the accuracy and comprehensiveness of our research. Your expert feedback has provided important guidance for our

work. In response, we have further supplemented the manuscript with additional clarifications and refined the schematic illustrations accordingly.

Fig. 1 Design and structure characterizations of MnBTC-Ru complex. **b** Natural Mn peroxidase-inspired construction of RuMn-oxygen complex for ROS biocatalysis.

Supplementary Fig. 7. Schematic illustration of the valence states of Mn under various coordination environments in MnBTC-Ru.

(3) Figure 2b shows the appearance of Mn^{4+} in MnBTC, which is somewhat unbelievable. It is rare for Mn^{2+} to directly change to Mn^{4+} in MOFs. Please verify that, in general, the Mn in MnBTC structure is Mn^{3+} , unless MnBTC is synthesized from Mn^{3+} as the raw material. However, this is clearly not the case in this article (it is $Mn(NO_3)_2$).

Response to comment:

Thank you for your thorough review of our manuscript and for the insightful suggestions you have provided. Your concerns regarding the valence state of Mn in MnBTC are particularly important and have prompted us to undertake a more comprehensive analysis of the electronic structure of Mn within the MnBTC-Ru. Below, we present a detailed response to this issue, along with the corresponding revisions made.

We thank the reviewer for your correct clarification that within conventional MOF structures, the direct oxidation of Mn^{2+} to Mn^{4+} is uncommon, with Mn^{3+} typically representing the stable valence state. After careful checking, we acknowledge that our previous peak fitting contained errors. According to the reviewer's suggestion, we have revised our analysis by re-fitting the XPS spectra using more rigorous constraints, including fixed peak position and half-peak widths. The revised results confirm a predominance of Mn^{3+} species within the MOF (Fig. 2b). The MnBTC-Ru system is inherently complex and results from multiple interacting factors that influence the valence state distribution of Mn: 1) In a typical MnO_6 coordination environment, where Mn is not bonded to Ru and is coordinated exclusively to the oxygen atoms of the BTC ligands, the Mn valence state remains stably at +3. 2) In cases of coordination unsaturation—where Mn is similarly not bonded to Ru—coordination defects result in a reduction of the Mn valence state to +2. 3) The electron transfer between the RuMn active centers—formed by the addition of Ru nanoclusters—induces Mn to exhibit valence states ranging from +2 to +3, where Mn^{2+} corresponds to electron gain and Mn^{3+} to the absence thereof. This combined effect of Mn-Ru bonding interactions and ligand coordination results in the coexistence of multiple Mn valence states (+2 and +3) within the system (Supplementary Fig. 7).

We sincerely appreciate your valuable input, which has greatly enhanced the accuracy of our data. We have accordingly incorporated the relevant discussion into the revised manuscript and cited pertinent literature on MOFs (*Adv. Mater.* **2024**, 36, 2206421; *Adv. Sci.* **2025**, 12, 2415477; *Adv. Healthcare Mater.* **2024**, 13, 2304141) to support the validity of this phenomenon, as also shown below:

Page 8 in the revised manuscript: “Mn 2p analysis reveals the coexistence of Mn^{2+} and Mn^{3+} valence states within MnBTC-Ru, with MnO_6 centers coordinated exclusively to the oxygen atoms of BTC ligands exhibiting a +3 oxidation state, Mn species with unsaturated coordination exhibiting a +2

oxidation state, and electron transfer between the RuMn active centers inducing Mn to display valence states ranging from +2 to +3 (Fig. 2b and Supplementary Fig. 7)⁴⁷⁻⁴⁹.”

Fig. 2 Analysis of precise atomic coordination structures in MnBTC-Ru complex. The high-resolution XPS of **b** Mn 2p for different catalysts.

Supplementary Fig. 7. Schematic illustration of the valence states of Mn under various coordination environments in MnBTC-Ru.

(4) How to achieve targeted therapy for tumors after introducing Mn ions in this study, and will there be stronger potential side effects?

Response to comment:

Thank you for your thorough review of our manuscript and for the insightful suggestions you have provided. Your concerns regarding how targeted tumor therapy is achieved following the introduction of Mn ions in this study, as well as the potential side effects, are particularly important. Below, we present a detailed response to this issue, along with the corresponding revisions made.

In tumor radiotherapy, the effectiveness of radiosensitizers depends on their ability to align closely with the localized and short-duration nature of radiation exposure. To maximize therapeutic benefit during the limited treatment window, these agents must rapidly concentrate within the tumor region. However, achieving high local drug accumulation while maintaining systemic safety remains a major challenge in nanomedicine. Intravenous delivery often falls short of achieving precise targeting within this narrow timeframe. In contrast, the intratumoral injection offers a more effective approach by improving local drug retention and minimizing systemic side effects.

The intratumoral injection therapy of MnBTC-Ru reported in this work draws inspiration from NBTXR3, the first FDA-approved hafnium oxide-based radiosensitizing nanomaterial, which also employs intratumoral injection to maximize local drug concentration during radiotherapy while minimizing systemic exposure. The clinical validation of NBTXR3 has demonstrated that intratumoral injection is not only feasible but effective for achieving localized radiosensitization (*Lancet Oncol.* **2019**, 20, 1148–1159; *Clin. Cancer. Res.* **2017**, 23, 908–917). This delivery approach ensures drug bioavailability precisely at the irradiation site and substantially reduces systemic toxicity, thereby enhancing overall biosafety.

Several prior studies have established the favorable biocompatibility of manganese-containing nanomaterials (*Sci. Adv.* **2024**, 10, eadk6610; *J. Nanobiotechnology* **2024**, 22, 664). In our study, the safety of RT+MnBTC-Ru treatment was evaluated through Hematoxylin and Eosin staining (H&E) staining of major organs and blood chemistry analysis. After treatment, H&E staining was performed on the heart, liver, spleen, lungs, and kidneys of the mice. The results show intact tissue structures and normal cellular morphology, with no evident pathological changes (Supplementary Fig. 18). Additionally, blood chemistry analysis confirmed that all measured parameters remained within the normal range, indicating no adverse effects on organ function (Supplementary Fig. 19). These results collectively demonstrate the favorable safety profile of RT+MnBTC-Ru treatment.

After confirming the potent antitumor efficacy and favorable safety profile of MnBTC-Ru, we further investigated its *in vivo* fate to elucidate the basis of its tumoricidal effects while maintaining minimal systemic toxicity. Quantitative analysis of Ru levels in tumors, major organs, and metabolic excreta was performed using inductively coupled plasma mass spectrometry (ICP-MS) (Supplementary Fig. 20). The results show that at all three-time points, the percentage of injected dose (%ID) of Ru in tumor tissue remained the highest, significantly exceeding that in other tissues, providing crucial support for its potent radiosensitizing and antitumor effects. In major organs, higher Ru accumulation was observed in the liver and kidneys, suggesting hepatic and renal involvement in MnBTC-Ru metabolism. ICP-MS analysis of urine and feces further confirmed that MnBTC-Ru is primarily excreted via hepatic and renal pathways. Notably, minimal Ru accumulation was detected in the heart, spleen, lungs, and blood, reinforcing the systemic safety of MnBTC-Ru. Furthermore, the reconstruction of CT imaging provides a spatial visualization of MnBTC-Ru in relation to the subcutaneous tumor (Fig. 6b).

In summary, the intratumoral injection approach used in this study successfully reduced systemic exposure of MnBTC-Ru to non-target organs. The experimental data confirm that MnBTC-Ru possesses outstanding biocompatibility and biodegradability.

Supplementary Fig. 18. Representative H&E staining of major organs from CT26 tumor-bearing mice after different treatments (scale bar = 100 μ m), experiments were repeated independently three times with similar results.

Supplementary Fig. 19. Results of blood chemistry parameters from mice after different treatments (n = 3 independent replicates). Blood chemistry parameters include white blood cell (WBC), red blood cell (RBC), platelet count (PLT), hemoglobin (HGB), alanine aminotransferase (ALT), aspartate aminotransferase (AST), creatinine (CREA), albumin (AIB), Creatine Kinase (CK). Source data are provided as a Source Data file.

Supplementary Fig. 20. Biodistribution of MnBTC-Ru in major organs and blood of mice at different time points after intratumoral injection of MnBTC-Ru (n = 3 independent replicates, data are presented as mean \pm SD). Source data are provided as a Source Data file.

(5) *In this study, there were 5 mice in each treatment group, and the data error was very small. However, the individual differences among mice are generally large. Can such a small error be made (Figure 5b, 5c, 5d)? Especially for the control group in Figure 5c, please provide a detailed explanation of the experimental and computational processes.*

Response to comment:

We sincerely thank the reviewers for their thoughtful comments and suggestions. In particular, we appreciate the observation regarding the potential impact of inter-individual variability among mice on our experimental results. Upon careful review of our data, we recognize that the original presentation may not have fully captured the differences in individual treatment responses. To address this, we have revised the manuscript to include detailed experimental protocols and raw data, providing a clearer presentation.

All animal experiments in this study were conducted using age- and weight-matched female BALB/c mice that sourced from the same batch and housed under specific pathogen-free (SPF)

conditions. Before initiating experiments, mice were acclimated for one week to ensure physiological stability. Tumor models were established by subcutaneously injecting an identical passage of cell suspension into the right flank of each mouse. Once tumors reached the target size, mice were randomly assigned to experimental groups for further analysis. Consistent with previous research reports (*Cell Metab.* **2024**, 36, 2493-2510e9; *Cancer Cell.* **2023**, 41, 272-287.e9), the sample size of five mice per group in our tumor xenograft model adheres to established research standards in this field and provides a reliable evaluation of treatment efficacy. As shown in Figure 5b, the average tumor growth curves are presented for each experimental group, while Figure 5c displays the individual tumor growth kinetics for every mouse, with corresponding raw data provided in Table R1. Furthermore, Figure 5d illustrates the mean tumor weights across groups, supported by detailed quantitative data in Table R2.

Fig. 5b Average tumor growth curves of CT26 tumor-bearing mice after different treatments (n = 5 biologically independent mice per group).

Fig. 5c Individual tumor growth kinetics of CT26 tumor-bearing mice after different treatments (n = 5 biologically independent mice per group).

Fig. 5d Tumor weight individual tumor growth kinetics of CT26 tumor-bearing mice after different treatments (n = 5 biologically independent mice per group).

Table R1 Longitudinal tumor volume measurements for each mouse in all groups

Time (days)	0	2	4	6	8	10	12
Tumor volume of control group (mm ³)	90.416250	332.412452	573.152254	847.598766	1092.62720	1534.54433	1656.49792
	93.087072	152.332259	261.013856	472.200552	968.657264	1241.40625	1864.73248
	86.6903775	140.564736	307.470948	563.388883	896.616171	1033.97065	1732.75451
	92.508318	138.354232	484.074920	850.732218	1113.06023	1370.2221	1857.08531
	97.388388	219.785825	461.384396	674.668896	918.067446	1254.20400	1319.29791

Mean	92.02	196.69	417.42	681.72	997.81	1287.87	1686.07
Standard Deviation	3.91	82.88	129.60	168.84	99.65	183.97	222.90
Tumor volume of RT group (mm ³)	93.323264	72.4771495	177.380313	165.908475	164.192384	175.242375	227.047374
	81.9819	179.87346	242.094258	172.443776	197.047058	279.82319	384.6864
	96.2329375	193.886496	240.157712	176.81125	212.595578	249.823013	303.098188
	80.417526	246.94285	273.86374	222.934761	237.580331	250.229808	261.139263
	82.337328	246.019968	300.507768	296.84835	281.984274	275.38	280.589971
Mean	86.86	187.84	246.80	206.99	218.68	246.10	291.31
Standard Deviation	7.34	71.22	46.10	55.05	44.27	41.97	59.188
Tumor volume of MnBTC-Ru group (mm ³)	80.713386	145.022202	292.552704	586.035712	784.47596	1006.56935	1296.56869
	80.691055	98.722944	214.856856	315.926826	347.656273	446.5615	559.289063
	83.380104	83.380104	83.380104	83.380104	83.380104	83.380104	83.380104
	93.6096	110.895624	264.424662	534.810078	651.362166	824.633136	943.562654
	89.170454	138.703263	275.799636	437.40446	497.967125	645.609536	888.299232
Mean	85.51	121.39	257.46	432.97	532.28	695.85	881.84
Standard Deviation	5.70	19.64	30.61	130.17	184.57	222.15	276.41
Tumor volume of RT+MnBTC-Ru group (mm ³)	80.4955595	127.505381	135.278322	166.90332	163.792657	158.195976	153.927816
	96.0553125	96.0553125	96.0553125	96.0553125	96.0553125	96.0553125	96.0553125
	96.2100055	168.567399	212.83116	168.36625	173.5866	177.510711	188.315631
	83.232549	42.328125	42.7591395	50.509644	50.866704	57.199338	66.668832
	79.3454535	20.817504	29.47772	20.829029	30.439714	41.6009925	49.609186
Mean	87.07	80.30	99.29	94.30	99.81	105.94	113.00
Standard Deviation	8.39	64.12	75.49	68.81	65.43	60.17	58.21

Table R2 Tumor weight measurements in different treatment groups.

group	control	RT	MnBTC-Ru	RT+MnBTC-Ru
Tumor weight (g)	2.20	0.70	1.10	0.15
	1.50	0.70	1.00	0.11
	1.30	0.60	0.90	0.09
	1.30	0.40	0.80	0.03
	1.10	0.30	0.50	0.01
Mean	1.48	0.54	0.86	0.08
Standard Deviation	0.43	0.18	0.23	0.06

(6) *This article conducted transcriptomic analysis, but why is there no detailed KEGG analysis? KEGG analysis can provide a clearer explanation for the cellular pathway changes caused by MnBTC and MnBTC-Ru.*

Response to comment:

We sincerely thank you for your valuable comments and suggestions, which have provided important guidance for the further development of our research. In response to your concerns regarding the KEGG analysis, we conducted two complementary approaches based on the existing sequencing data to elucidate the alterations in cellular pathways more comprehensively.

First, we performed KEGG enrichment analysis based on differentially expressed genes (DEGs). This method allows for the rapid identification of biological pathways that may be closely associated with the observed phenotype by directly analyzing the enriched pathways of DEGs. The results revealed that the RT+MnBTC-Ru combination treatment significantly activated multiple key pathways involved in redox regulation, immune responses, and inflammation-apoptosis signaling. Specifically, the enrichment of oxidative phosphorylation and HIF-1 signaling pathways suggests metabolic reprogramming in tumor cells, potentially accompanied by disruptions in redox homeostasis and altered hypoxic response mechanisms (*Nat. Rev. Cancer* **2003**, 3, 721–732; *Cell Metab.* **2021**, 33, 2398–2414). In addition, the enrichment of cytokine–cytokine receptor interaction and T cell receptor signaling pathways reflects molecular features of immune activation. Meanwhile, the significant enrichment of the NF- κ B and TNF signaling pathways suggests the potential co-activation of inflammatory responses and pro-apoptotic signaling under this treatment (*Nat. Rev. Immunol.* **2023**, 23, 289–303). These findings suggest that MnBTC-Ru may enhance tumor cell sensitivity to radiotherapy by coordinately modulating oxidative stress, immune activation, and cell death signaling pathways. Specifically, it disrupts ROS homeostasis to induce sustained oxidative damage, activates cytokine-T cell signaling cascades to boost antitumor immunity, and remodels the NF- κ B/TNF signaling axis to promote a death-prone cellular phenotype. This synergistic activation of multiple pathways ultimately leads to programmed cell death (Supplementary Fig. 25).

We next performed KEGG pathway enrichment analysis based on the results of Mfuzz clustering (Fig. R5 and R6). Gene expression profiles from each treatment group were subjected to Mfuzz soft

clustering, and the resulting gene clusters were individually analyzed for pathway enrichment. This approach not only identifies significantly enriched pathways but also captures condition-specific expression dynamics across different treatment groups. In the RT+MnBTC group, genes associated with apoptosis, necrosis, and other cell death-related pathways were markedly upregulated. In parallel, immune- and inflammation-related pathways—including cytokine–cytokine receptor interaction, NF- κ B signaling, and TNF signaling—also showed persistent activation. These findings extend traditional static KEGG analysis by uncovering condition-specific expression dynamics that contribute to the understanding of the molecular mechanisms underlying RT+MnBTC treatment.

By combining these two analytical approaches, we not only cross-validated the identified pathways but also systematically captured their dynamic expression patterns under different treatment conditions. This integrative strategy provides more robust support for our conclusions and offers deeper insights into the underlying biological mechanisms. We sincerely appreciate your valuable suggestions.

Page 20 in the revised manuscript: “In addition, pathway enrichment analysis based on the Kyoto Encyclopedia of Genes and Genomes (KEGG) database (Supplementary Fig. 25) revealed pronounced activation of signaling pathways in the RT+MnBTC combination treatment group, predominantly associated with redox regulation, immune modulation, and inflammation-apoptosis signaling. Moreover, the gene set enrichment analysis (GSEA) result shows activated pathways, such as immune response and immune system process, underscoring the pivotal role of RT+MnBTC-Ru in the activation of antitumor immunity (Fig. 6g). These findings suggest potential synergistic effects between RT+MnBTC-Ru and checkpoint inhibitors, presenting a new therapeutic avenue for cancer radiosensitization treatment.”

KEGG Enrichment Scatter Plot

Supplementary Fig. 25. KEGG enrichment analysis of upregulated genes in the RT+MnBTC-Ru group. Source data are provided as a Source Data file.

Fig. R5 Mfuzz clustering analysis of gene expression patterns (Clusters 1-4). Mfuzz soft clustering reveals treatment-induced differential expression dynamics (Clusters 1-4). Each cluster represents a gene set with similar expression patterns, visualized by characteristic expression trend curves. The y-axis indicates standardized expression values (z-scores), while the x-axis represents different experimental treatment groups. Thick lines denote cluster centroid trendlines, with shaded areas reflecting the membership degree of individual genes within each cluster.

Fig. R6 Mfuzz clustering analysis of gene expression patterns (Clusters 5-8). Mfuzz soft clustering reveals treatment-induced differential expression dynamics (Clusters 5-8). Each cluster represents a gene set with similar expression patterns, visualized by characteristic expression trend curves. The y-axis indicates standardized expression values (z-scores), while the x-axis represents different experimental treatment groups. Thick lines denote cluster centroid trendlines, with shaded areas reflecting the membership degree of individual genes within each cluster.

(7) *The proportion of ROS produced by each component should be calculated, indicating which component plays a key role and through which signaling pathway it acts on cells.*

Response to comment:

We gratefully acknowledge you for your insightful comments. In response to the recommendation to clarify the specific functions of each component in ROS production, we have conducted a thorough investigation using systematic *in vitro* enzyme-mimetic assays, which employed the o-phenylenediamine colorimetric method, along with complementary *in vitro* cellular experiments.

1) Enzyme-like assays: The experimental results demonstrate that the ROS production activity of individual MnBTC or Ru clusters (C-Ru) is markedly lower than that observed in the MnBTC-Ru-based enzyme-mimicking system. Moreover, following radiation treatment, the ROS generation capacity of MnBTC-Ru is significantly enhanced, whereas no comparable enhancement is detected for MnBTC or C-Ru alone. These findings provide compelling evidence that the advantages of the Ru-Mn synergistic center, along with its cooperative interaction with radiotherapy, constitute the critical mechanism driving increased ROS production; both elements are essential for this mechanism. It is important to note, however, that due to the absence of a cellular microenvironment, the radiation-enhancing effect observed *in vitro* is relatively constrained.

2) *In vitro* cellular experiments: This synergistic effect of MnBTC-Ru and radiotherapy was further validated through a series of *in vitro* experiments. First, using the ROS-sensitive probe DCFH-DA, we quantify intracellular ROS levels across treatment groups and find that cells receiving RT+MnBTC-Ru generate significantly higher ROS, demonstrating that the combination therapy effectively promotes oxidative stress (Fig. 4b). Recognizing that elevated ROS can inflict damage on critical cellular components—most notably DNA—we then examined DNA integrity by assessing γ -H2AX expression, a well-established marker of DNA double-strand breaks (Supplementary Fig. 11). The pronounced increase of γ -H2AX signals in the RT+MnBTC-Ru group confirmed that ROS accumulation leads to substantial DNA damage. Building on this, we investigated the downstream cellular response by measuring apoptosis through flow cytometry (Supplementary Fig. 12). Consistent with the DNA damage findings, the RT+MnBTC-Ru treatment induced the highest apoptosis levels, thereby substantiating the mechanism whereby ROS-mediated DNA damage triggers cell apoptosis. Together, these experiments establish a coherent sequence linking enhanced ROS production to DNA

damage and, ultimately, to apoptosis induction.

3) RNA-seq and bioinformatics analysis. To elucidate the molecular mechanisms underlying the effects of RT+MnBTC-Ru combination therapy on tumor cells, we have performed an integrated analysis combining Mfuzz clustering and KEGG pathway enrichment based on transcriptomic data from *in vivo* tumor tissues (Fig. R5 and R6). This approach enables a systematic characterization of the dynamic gene expression profiles under different treatment conditions. Specifically, Mfuzz cluster analysis reveals distinct expression profiles of gene clusters across treatment groups, while KEGG analysis identified significantly enriched biological pathways associated with these genes. The results demonstrate that RT+MnBTC-Ru treatment markedly upregulated genes involved in apoptosis, necrosis, and other cell death pathways, indicating the effective activation of tumor cell death programs by this combination therapy. Notably, key genes associated with inflammatory responses and immune-related pathways (cytokine–cytokine receptor interaction, NF- κ B signaling, and TNF signaling pathway) also showed significant upregulation in the RT+MnBTC-Ru group, suggesting that this therapeutic strategy exerts its antitumor effects through multiple synergistic mechanisms.

In conclusion, the MnBTC-Ru-based catalytic system exhibits enhanced enzyme-mimetic activity under radiotherapy, thus promoting ROS generation and leading to irreversible DNA damage and apoptosis. The resulting cell death releases immunostimulatory signals, potentially amplifying systemic antitumor immune responses. Together, these findings establish a strong rationale for using MnBTC-Ru as a dual-function platform for radiosensitization and immune activation in cancer therapy. Detailed experimental procedures, comprehensive detection data, and pertinent discussions have been incorporated into the revised manuscript, as also shown below:

Page 11 in the revised manuscript: “Subsequent use of electron paramagnetic resonance (EPR) spectroscopy, **the o-phenylenediamine (OPD) colorimetric assay** reveals the enhanced ROS production by the biocatalytic MnBTC-Ru complex in the presence of X-ray radiation (Fig. 3i, j **and Supplementary Fig. 13**). **The Ru-Mn synergistic center, together with its cooperative interaction with radiotherapy, constitutes the principal mechanism driving ROS generation.**”

Supplementary Fig. 13. The proportion of ROS produced by each component is detected by o-phenylenediamine (OPD) ($n = 6$ independent experiments, data are presented as mean \pm SD). Statistical significance was assessed was calculated using two-tailed Student's t-test, all tests were two-sided. Source data are provided as a Source Data file.

Page 22 in the revised supporting information: “**X-ray-activated enzyme-like activity.** 10 mg of o-phenylenediamine (OPD) was dissolved in 1 mL of DMF. Then, 10 μ L of H₂O₂ (0.1 M), 24 μ L of OPD (10 mg/mL), and 20 μ L of different materials (4 mg/mL) were successively added to 1925 μ L of NaOAc/HOAc buffer (100 mM, pH 5.5). 200 μ L of the resulting solution were transferred into a 96-well plate. After irradiation with X-rays at 6 Gy, the absorbance was measured at the maximum absorption wavelength of 448 nm.”

Fig. 4b The representative fluorescence images and c quantitative flow cytometry show the ROS generation in CT26 cells in different groups (n = 3 independent replicates; scale bar = 100 μ m).

Supplementary Fig. 14. γ -H2AX immunofluorescence assay. γ -H2AX assays showing DNA double-strand breaks in CT26 cells treated with PBS, MnBTC, or MnBTC-Ru and X-ray irradiation. Scale bar = 100 μ m. Experiments were repeated independently three times with similar results.

Supplementary Fig. 15. The Annexin V/PI analysis of CT26 cells in different groups. The graph showed the percentage of apoptotic cells (early apoptotic, late apoptotic) in different groups (n = 3 independent replicates). Source data are provided as a Source Data file.

Fig. R5 Mfuzz clustering analysis of gene expression patterns (Clusters 1-4). Mfuzz soft clustering reveals treatment-induced differential expression dynamics (Clusters 1-4). Each cluster represents a gene set with similar expression patterns, visualized by characteristic expression trend curves. The y-axis indicates standardized expression values (z-scores), while the x-axis represents different experimental treatment groups. Thick lines denote cluster centroid trendlines, with shaded areas reflecting the membership degree of individual genes within each cluster.

Fig. R6 Mfuzz clustering analysis of gene expression patterns (Clusters 5-8). Mfuzz soft clustering reveals treatment-induced differential expression dynamics (Clusters 5-8). Each cluster represents a gene set with similar expression patterns, visualized by characteristic expression trend curves. The y-axis indicates standardized expression values (z-scores), while the x-axis represents different experimental treatment groups. Thick lines denote cluster centroid trendlines, with shaded areas reflecting the membership degree of individual genes within each cluster.

We thank all referees again for their helpful comments and suggestions, and hope that this significantly revised manuscript is now acceptable for publication in *Nature Communications*.

Best Regards,

Yours Sincerely,

Prof. Dr. Chong Cheng (on behalf of the authors)